# SEABO: A Simple Search-Based Method for Offline Imitation Learning

**Jiafei Lyu**[1] [*]**Xiaoteng Ma**[2]**, Le Wan**[3]**, Runze Liu**[1]**, Xiu Li**[1,†]**, Zongqing Lu**[4,†]

[1]Tsinghua Shenzhen International Graduate School, Tsinghua University
[2]Department of Automation, Tsinghua University, [3]IEG, Tencent
[4]School of Computer Science, Peking University
`lvjf20@mails.tsinghua.edu.cn, li.xiu@sz.tsinghua.edu.cn, zongqing.lu@pku.edu.cn`

## ABSTRACT

Offline reinforcement learning (RL) has attracted much attention due to its ability in learning from static offline datasets and eliminating the need of interacting with the environment. Nevertheless, the success of offline RL relies heavily on the offline transitions annotated with reward labels. In practice, we often need to hand-craft the reward function, which is sometimes difficult, labor-intensive, or inefficient. To tackle this challenge, we set our focus on the offline imitation learning (IL) setting, and aim at getting a reward function based on the expert data and unlabeled data. To that end, we propose a simple yet effective search-based offline IL method, tagged SEABO. SEABO allocates a larger reward to the transition that is close to its closest neighbor in the expert demonstration, and a smaller reward otherwise, all in an unsupervised learning manner. Experimental results on a variety of D4RL datasets indicate that SEABO can achieve competitive performance to offline RL algorithms with ground-truth rewards, given only a single expert trajectory, and can outperform prior reward learning and offline IL methods across many tasks. Moreover, we demonstrate that SEABO also works well if the expert demonstrations contain only observations. Our code is publicly available at https://github.com/dmksjfl/SEABO.

## 1 INTRODUCTION

In recent years, reinforcement learning (RL) (Sutton & Barto, 2018) has made prominent achievements in fields like video games (Mnih et al., 2015; Schrittwieser et al., 2020), robotics (Kober et al., 2013), nuclear fusion control (Degrave et al., 2022), etc. It is known that RL is a reward-oriented learning paradigm. Online RL algorithms typically require an interactive environment for data collection and improve the policy through trial-and-error. However, continual online interactions are usually expensive, time-consuming, or even dangerous in many practical applications. Offline RL (Lange et al., 2012; Levine et al., 2020), instead, resorts to learning optimal policies from previously gathered datasets, which are composed of trajectories containing observations, actions, and rewards.

A bare fact is that reward engineering is often difficult, expensive, and labor-intensive. It is also hard to specify or abstract a good reward function given some rules. To overcome this challenge in the *offline* setting, there are generally two methods. First, one can train the policy via the behavior cloning (BC) algorithm (Pomerleau, 1988), but its performance is heavily determined by the performance of the data-collecting policy (*a.k.a.*, the behavior policy). Second, one can learn a reward function from some expert demonstrations and assign rewards to the unlabeled data in the dataset. Then, the policy can be optimized by leveraging the reward. This is also known as *offline imitation learning* (offline IL). Note that in many real-world tasks, acquiring a few expert demonstrations is easy (*e.g.*, ask a human expert to operate the system) and affordable.

However, it turns out that, similar to offline RL, offline IL also suffers from distribution shift issue (Kim et al., 2022b; DeMoss et al., 2023), where the learned policy deviates from the data-collecting policy, leading to poor performance during evaluation. Prior works concerning on *distribution cor-*

---

[*]Work done while working as an intern at Tencent. [†] Corresponding Authors.

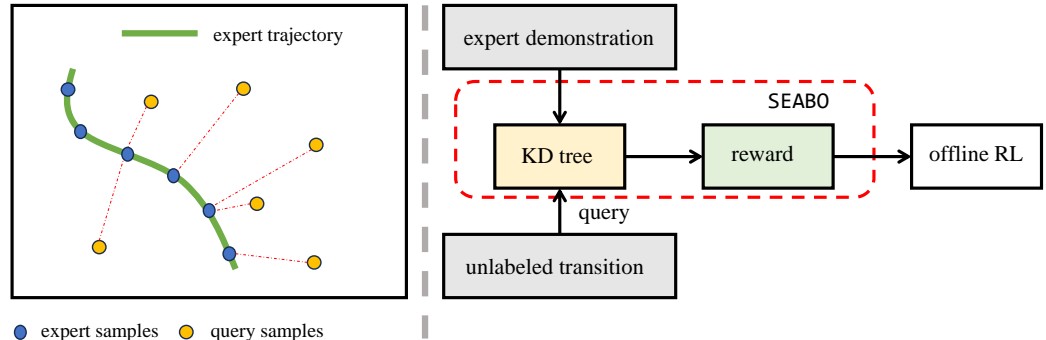

Figure 1: **Left:** The key idea behind SEABO. We assign larger rewards to transitions that are closer to the expert demonstration, and smaller rewards otherwise. The dotted lines connect the query samples with their nearest neighbors along the demonstration. **Right:** Illustration of the SEABO framework. Given an expert demonstration, we first construct a KD-tree and then feed the unlabeled samples into the tree to query their nearest neighbors. We use the resulting distance to calculate the reward label. Then one can adopt any existing offline RL algorithm to train on the labeled dataset.

*rection estimation* (DICE family) address this by enforcing the learned policy to be close to the behavior policy via a distribution divergence measure (*e.g.*, $f$-divergence (Ghasemipour et al., 2019; Ke et al., 2019)). However, such distribution matching schemes can incur training instability (Ma et al., 2022) and over-conservatism (Yu et al., 2023), and they often involve training task-specific discriminators. On the other hand, some works seek to decouple the processes of reward annotation and policy optimization (Zolna et al., 2020; Luo et al., 2023). However, they involve solving complex optimal transport problems or contrasting expert states and unlabeled trajectory states.

In this paper, we propose a simple yet effective alternative, **SEA**rch-**B**ased method for **O**ffline imitation learning, namely SEABO, that leverages search algorithms to acquire reward signals in an *unsupervised learning manner*. As illustrated in Figure 1 (left), we hypothesize that the transition is near-optimal if it lies close to the expert trajectory, hence larger reward ought to be assigned to it, and vice versa. To that end, we propose to determine whether the sample approaches the expert trajectory via measuring the distance between the query sample and its nearest neighbor in the expert trajectory. In practice, as depicted in Figure 1 (right), SEABO first builds a KD-tree upon expert demonstrations. Then for each unlabeled sample in the dataset, we query the tree to find its nearest neighbor, and measure their distance. If the distance is small (*i.e.*, close to expert trajectory), a large reward will be annotated, while if the distance is large (*i.e.*, stray away from the expert trajectory), the assigned reward is low. SEABO is efficient and easy to implement. It can be combined with any existing offline RL algorithm to acquire a meaningful policy from the static offline dataset.

Empirical results on the D4RL (Fu et al., 2020) datasets show that SEABO can enable the offline RL algorithm to achieve competitive or even better performance against its performance under ground-truth rewards with *only one expert trajectory*. SEABO also beats recent strong reward annotation methods and imitation learning baselines on many datasets. Furthermore, we also demonstrate that SEABO can learn effectively when the expert demonstrations are composed of pure observations.

## 2 PRELIMINARY

We formulate the interaction between the environment and policy as a Markov Decision Process (MDP) specified by the tuple $\langle \mathcal{S}, \mathcal{A}, p, r, \gamma, p_0 \rangle$, where $\mathcal{S}$ is the state space, $\mathcal{A}$ is the action space, $p : \mathcal{S} \times \mathcal{A} \mapsto \mathcal{S}$ is the transition dynamics, $r : \mathcal{S} \times \mathcal{A} \mapsto \mathbb{R}$ is the scalar reward signal, $\gamma \in [0, 1]$ is the discount factor, $p_0$ is the initial state distribution. A policy $\pi(a|s)$ outputs the action based on the state $s$. We assume that the underlying MDP has a finite horizon. The goal of RL is to maximize the discounted future reward: $J(\pi) = \mathbb{E}_{s_0 \sim p_0} \mathbb{E}_{a \sim \pi, s_{t+1} \sim p(\cdot|s_t, a_t)} [\sum_{t=0}^{T-1} \gamma^t r(s_t, a_t)]$. Whereas, in many scenarios, it is often hard for one to get the reward signals. It is more common that the *unlabeled trajectory*, $\tau = \{s_0, a_0, \ldots, s_t, a_t, \ldots, s_T\}$, is collected. This poses veritable challenges for applying offline RL algorithms.

In this paper, we focus on the offline IL setting. We assume that we have access to the dataset of expert demonstrations $\mathcal{D}_e = \{\tau_e^{(i)}\}_{i=1}^M$, and a dataset of unlabeled data $\mathcal{D}_u = \{\tau_u^{(i)}\}_{i=1}^N$, where $M$ and $N$ are the sizes of the expert dataset and unlabeled dataset, respectively. The unlabeled trajectories are gathered by some unknown behavior policy $\mu$. Note that we allow the expert demonstrations to either contain actions or do not contain actions. We aim at attaining the reward function by extracting information from the expert trajectories and unlabeled trajectories, and assigning rewards to the unlabeled datasets, without any interactions with the environment. Then we can train the policy using any offline RL algorithm.

## 3 RELATED WORK

**Offline Reinforcement Learning.** In offline RL (Lange et al., 2012; Levine et al., 2020), the agent is not permitted to interact with the environment, and can only learn policies from previously gathered dataset $\mathcal{D} = \{(s_i, a_i, r_i, s_{i+1})\}_{i=1}^N$, where $N$ is the dataset size. Existing work on offline RL can be generally categorized into model-based (Yu et al., 2020; 2021; Kidambi et al., 2020; Lyu et al., 2022b; Rigter et al., 2022; Lu et al., 2022a; Chen et al., 2021; Janner et al., 2021; Uehara & Sun, 2022; Zhang et al., 2023) and model-free approaches (Fujimoto et al., 2019; Fujimoto & Gu, 2021; Kumar et al., 2020; Kostrikov et al., 2022; Lyu et al., 2022c;a; Cheng et al., 2022; Zhou et al., 2020; Ran et al., 2023; Bai et al., 2022; Yang et al., 2024). The success of these methods rely heavily on the requirement that the datasets must contain annotated reward signals.

**Imitation Learning.** Imitation Learning (IL) considers optimizing the behavior of the agent given some expert demonstrations, and no reward is needed. The primary goal of IL is to mimic the behavior of the expert demonstrator. Behavior cloning (BC) (Pomerleau, 1988) directly performs supervised regression or maximum-likelihood on expert demonstrations. Yet, BC can suffer from compounding error and may result in performance collapse upon unseen states (Ross et al., 2011). Another line of work, inverse reinforcement learning (IRL) (Arora & Doshi, 2021), first learns a reward function using expert demonstrations, and then utilizes this reward function to train policies with RL algorithms. Typical IRL algorithms include adversarial methods (Ho & Ermon, 2016; Jeon et al., 2018; Kostrikov et al., 2019; Baram et al., 2017), maximum-entropy approaches (Ziebart et al., 2008; Boularias et al., 2011), normalizing flows (Freund et al., 2023), etc. However, these methods often require abundant online transitions to train a good policy. Imitation learning without online interactions, which is the focus of our work, is hence attractive and remains an active area. There are many advances in offline IL, such as applying online IRL algorithms in the offline setting (Zolna et al., 2020; Yue et al., 2023), using energy-based methods (Jarrett et al., 2020), weighting the BC loss with the output of the trained discriminator (Xu et al., 2022), etc. Among them, DICE (Nachum et al., 2019) family receives much attention. Methods like ValueDICE (Kostrikov et al., 2020), DemoDICE (Kim et al., 2022b), and LobsDICE (Kim et al., 2022a) can consistently drub BC in the offline setting. Notably, a recent work, OTR (Luo et al., 2023), acquires the reward function in the offline setting via optimal transport. OTR decouples the processes of reward learning and policy optimization. Still, OTR needs to solve complex optimal transport problems. We, instead, explore to get the reward function via a search-based method.

**Search Algorithms.** Search algorithms (Korf, 1999) are critical components in artificial intelligence. Typical search algorithms include brute-force search algorithms (Dijkstra, 1959; Stickel & Tyson, 1985; Korf, 1985; Taylor & Korf, 1993), heuristic search approaches (Doran & Michie, 1966; Hart et al., 1968; Pohl, 1970; Edelkamp & Schrödl, 2011), etc. In this paper, we resort to the simple search approach, KD-tree (Bentley, 1975), for capturing the nearest neighbors of the unlabeled data in the expert demonstrations.

## 4 OFFLINE IMITATION LEARNING VIA SEARCH-BASED METHOD

In this section, we formally present our novel approach for offline imitation learning, **SEA**rch-**B**ased **O**ffline imitation learning (SEABO). We begin by analyzing the common formulation adopted in distribution matching IL methods (Ho & Ermon, 2016; Kim et al., 2020; Kostrikov et al., 2020), which attempt to match the state-action distribution of the agent $p_\pi$ and the expert $p_e$, often by means of minimizing some $f$-divergence measurement $D_f$: $\arg\min_\pi D_f(p_\pi \| p_e)$. Though these methods have promising results, they usually require training task-specific discriminators and suffer

from training instability (Wang et al., 2020; Ma et al., 2022). A natural question arises, can we get the reward signals without training neural networks?

Instead of measuring the distribution of states or state-action pairs, we want to determine the optimality of a single transition. Our idea is quite straightforward, the closer the transition is to the expert trajectory, the more optimal this transition is. The agent ought to pay more attention to those optimal transitions. This motivates us to measure how close the unlabeled transition is to the expert trajectories. We propose to achieve this by *finding the nearest neighbor of the query transition in the expert demonstrations*, and then measuring their distance (*e.g.*, Euclidean distance). If the distance is large, then the transition is away from the expert demonstration. While if the distance is small, it indicates that the transition is near-optimal, or even is exact expert data if the distance approaches 0. Intuitively, this distance can be interpreted as a reward signal.

To that end, we construct a function dubbed `NearestNeighbor(demo, query_sample)` that returns the nearest neighbor of the query sample in the expert demonstrations. Suppose the expert trajectories are made up of state-action pairs, then for the query sample $(s, a, s')$, we have:

$$(\tilde{s}_e, \tilde{a}_e, \tilde{s}'_e) = \texttt{NearestNeighbor}(\mathcal{D}_e, (s, a, s')). \tag{1}$$

Then we measure their deviation using some distance measurement $D$:

$$d = D((\tilde{s}_e, \tilde{a}_e, \tilde{s}'_e), (s, a, s')). \tag{2}$$

Afterward, following prior work (Cohen et al., 2022; Freund et al., 2023; Dadashi et al., 2021; Luo et al., 2023), we get the rewards via a squashing function: $r = \alpha \exp(-\beta \times d)$, where $\alpha$ and $\beta$ are hyperparameters that control the scale of the reward and the impact of the distance, respectively.

---

**Algorithm 1** SEArch-Based Offline Imitation Learning (SEABO)

---

1: **Require:** expert demonstrations $\mathcal{D}_e$, unlabeled dataset $\mathcal{D}_u$
2: Initialize $\mathcal{D}_{\text{label}} \leftarrow \emptyset$. Given distance measurement $D$
3: **for** $(s, a, s')$ in $\mathcal{D}_u$ **do**
4:     Find its nearest neighbor, $(\tilde{s}_e, \tilde{a}_e, \tilde{s}'_e) = \texttt{NearestNeighbor}(\mathcal{D}_e, (s, a, s'))$
5:     Measure the distance: $d = D((\tilde{s}_e, \tilde{a}_e, \tilde{s}'_e), (s, a, s'))$
6:     Get the reward signal via Equation 3
7:     $\mathcal{D}_{\text{label}} \leftarrow \mathcal{D}_{\text{label}} \cup (s, a, r, s')$
8: **end for**

---

We name the resulting method SEABO, and list its pseudo-code in Algorithm 1. For practical implementation of SEABO, we leverage KD-tree (Bentley, 1975) for searching the nearest neighbors, and adopt Euclidean distance (Torabi et al., 2019) as the distance measurement for simplicity (*i.e.*, the default setting of KD-tree). We also slightly modify the aforementioned formula of the reward function to make it better adapt to different tasks with one set of hyperparameters, which gives:

$$r = \alpha \exp\left(-\frac{\beta \times d}{|\mathcal{A}|}\right), \tag{3}$$

where $|\mathcal{A}|$ is the dimension of the action space. Note that this technique is also adopted in Dadashi et al. (2021). We choose to use $(s, a, s')$ to query since the magnitude of states and actions may be different. One possible solution is to query the demonstrations via $(\xi \times s, a), \xi \in \mathbb{R}^+$, but it introduces an additional hyperparameter that may need to be tuned per dataset. We empirically find that involving $s'$ in the query sample can ensure good performance across many tasks. The above procedure (as specified in Figure 1 (right)) also applies when the expert demonstrations contain only observations, because it is feasible that we find the nearest neighbors using only observations.

SEABO enjoys many advantages over prior reward learning methods or offline imitation learning algorithms. First, SEABO does not require any additional processing upon the offline dataset[1]. The unlabeled dataset can have different trajectory lengths, and the unlabeled trajectories can be fragmented, or even scattered, since SEABO computes the rewards only using the single transition instead of the entire trajectory. Second, SEABO does not require training reward models or discriminators, hence getting rid of the issues of training instability and hyperparameter tuning of the neural networks. Third, SEABO is easy to implement and can be combined with any offline RL algorithm.

---

[1] Methods like OTR (Luo et al., 2023) need zero padding of the unlabeled trajectories to ensure that they have identical length as the expert trajectories.

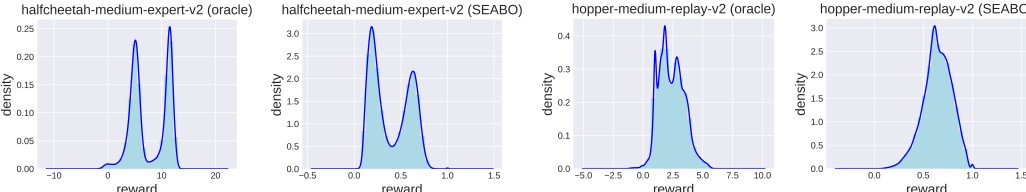

Figure 2: **Density plots of ground-truth rewards and rewards acquired by SEABO**. Note that oracle indicates the ground-truth rewards are plotted.

To show the effectiveness of our method, we plot the distribution of ground-truth rewards (oracle) and rewards given by SEABO. We choose two datasets, `halfcheetah-medium-expert-v2` and `hopper-medium-v2` from D4RL (Fu et al., 2020) as examples, and use $\alpha = 1, \beta = 0.5$, which is the same as our hyperparameter setup in Section 5. The results are summarized in Figure 2. We find that the reward distributions of SEABO resemble those of oracle. Notably, SEABO successfully gives two peaks in `halfcheetah-medium-expert`, indicating that it can distinguish samples of different qualities. These reveal that SEABO can serve as a good reward labeler, which validates its combination with off-the-shelf offline RL algorithms.

## 5  EXPERIMENTS

In this section, we empirically evaluate SEABO on D4RL datasets. We are targeted at examining, given only one single expert demonstration, whether SEABO can make different base offline RL algorithms recover or beat their performance with ground-truth rewards across varied tasks. We are also interested in exploring how SEABO competes against prior reward learning and offline imitation learning methods. We further investigate whether SEABO can work well if the expert demonstrations are composed of pure observations. Moreover, we check how different choices of search algorithms affect the performance of SEABO.

We discard reward signals in the D4RL datasets to form unlabeled datasets. For expert demonstrations, we follow Luo et al. (2023) and utilize the trajectory with the highest return in the raw dataset for ease of evaluation. One can also use a separate expert trajectory. All of the experiments in this paper are run for 1M gradient steps over five different random seeds, and the results are averaged at the final gradient step. We report the mean performance in conjunction with the corresponding standard deviation. We adopt the same squashing function for tasks under the same domain. Unless specified, we use the number of expert demonstrations $K = 1$ for evaluation. It is worth noting that SEABO is computationally efficient since there is only a single expert trajectory, and the time complexity of KD-tree gives $\mathcal{O}(d_f \log |\mathcal{D}_e|)$, where $d_f$ is the feature dimension size. It takes SEABO about 1 minute to annotate 1 million unlabeled transitions using merely CPUs. Hence, we believe the overall computation overhead from SEABO is minor and tolerable. We defer the experimental details and hyperparameter setup for all of our experiments to Appendix A.

### 5.1  MAIN RESULTS

**SEABO upon different base algorithms.** We first explore whether SEABO can aid different offline RL algorithms. We verify this by incorporating SEABO with two popular offline RL algorithms, TD3_BC (Fujimoto & Gu, 2021) and IQL (Kostrikov et al., 2022). We conduct experiments on 9 medium-level (`medium`, `medium-replay`, `medium-expert`) D4RL MuJoCo locomotion "-v2" datasets (`halfcheetah`, `hopper`, `walker2d`) and summarize the results in Table 1. One can see that IQL+SEABO beats IQL with ground-truth rewards on 6 out of 9 datasets, and TD3_BC+SEABO outperforms TD3_BC with raw rewards on 5 out of 9 datasets. On other datasets, SEABO can achieve competitive performance against the oracle performance. The overall scores of SEABO with IQL and TD3_BC exceed those of ground-truth rewards. This evidence indicates that SEABO can generate high-quality rewards and benefit different offline RL algorithms.

**SEABO competes against baselines.** To better illustrate the effectiveness of SEABO, we compare IQL+SEABO against the following strong reward learning and offline IL baselines: **ORIL** (Zolna et al., 2020), which learns the reward function by contrasting the expert demonstrations with the

Table 1: **Results of SEABO upon different base algorithms**. $\mu_{\max}$ denotes the normalized return of the *highest* return trajectory in the specific dataset, IQL and TD3_BC indicate that they are trained upon the ground-truth reward labels, while +*SEABO* indicates the algorithm is trained on the reward signals provided by SEABO. The normalized average scores at the final 10 episodes of evaluations are reported, along with standard deviations. We **bold** the mean score and highlight the cell if SEABO outperforms algorithms trained on ground-truth rewards.

| Task Name | $\mu_{\max}$ | IQL | IQL+SEABO | TD3_BC | TD3_BC+SEABO |
|---|---|---|---|---|---|
| halfcheetah-medium | 45.0 | 47.4±0.2 | 44.8±0.3 | 48.0±0.7 | 45.9±0.3 |
| hopper-medium | 99.5 | 66.2±5.7 | **80.9**±3.2 | 60.7±12.5 | **76.1**±4.2 |
| walker2d-medium | 92.0 | 78.3±8.7 | **80.9**±0.6 | 83.7±5.3 | 76.6±0.4 |
| halfcheetah-medium-replay | 42.4 | 44.2±1.2 | 42.3±0.1 | 44.4±0.8 | 43.0±0.4 |
| hopper-medium-replay | 98.6 | 94.7±8.6 | 92.7±2.9 | 64.8±25.5 | **96.3**±3.0 |
| walker2d-medium-replay | 89.9 | 73.8±7.1 | **74.0**±2.7 | 87.4±8.4 | 73.1±2.2 |
| halfcheetah-medium-expert | 92.8 | 86.7±5.3 | **89.3**±2.5 | 93.5±2.0 | **95.7**±0.4 |
| hopper-medium-expert | 116.0 | 91.5±14.3 | **97.5**±5.8 | 100.2±20.0 | **107.1**±3.3 |
| walker2d-medium-expert | 109.0 | 109.6±1.0 | **110.9**±0.2 | 109.5±0.5 | **109.7**±0.2 |
| Total Score | 785.2 | 692.4 | **713.3** | 692.3 | **723.5** |

Table 2: **Comparison of SEABO against some recent baselines.** We report the mean normalized scores and the corresponding standard deviations. We bold and highlight the mean score cell if it is close to or beats IQL trained on the raw rewards.

| Task Name | BC | 10%BC | IQL | IQL+ORIL | IQL+UDS | IQL+OTR | IQL+SEABO |
|---|---|---|---|---|---|---|---|
| halfcheetah-medium | 42.6 | 42.5 | 47.4±0.2 | **49.0**±0.2 | 42.4±0.3 | 43.2±0.2 | 44.8±0.3 |
| hopper-medium | 52.9 | 56.9 | 66.2±5.7 | 47.0±4.0 | 54.5±3.0 | **74.2**±5.1 | **80.9**±3.2 |
| walker2d-medium | 75.3 | 75.0 | 78.3±8.7 | 61.9±6.6 | 68.9±6.2 | **78.7**±2.2 | **80.9**±0.6 |
| halfcheetah-medium-replay | 36.6 | 40.6 | 44.2±1.2 | **44.1**±0.6 | 37.9±2.4 | 41.8±0.3 | 42.3±0.1 |
| hopper-medium-replay | 18.1 | 75.9 | 94.7±8.6 | 82.4±1.7 | 49.3±22.7 | 85.4±0.8 | **92.7**±2.9 |
| walker2d-medium-replay | 26.0 | 62.5 | 73.8±7.1 | **76.3**±4.9 | 17.7±9.6 | 67.2±6.0 | **74.0**±2.7 |
| halfcheetah-medium-expert | 55.2 | 92.9 | 86.7±5.3 | **87.5**±3.9 | 63.0±5.7 | 87.4±4.4 | **89.3**±2.5 |
| hopper-medium-expert | 52.5 | 110.9 | 91.5±14.3 | 29.7±22.2 | 53.9±2.5 | 88.4±12.6 | **97.5**±5.8 |
| walker2d-medium-expert | 107.5 | 109.0 | 109.6±1.0 | **110.6**±0.6 | 107.5±1.7 | **109.5**±0.3 | **110.9**±0.2 |
| Total Score | 466.7 | 666.2 | 692.4 | 588.5 | 495.1 | 675.8 | **713.3** |

unlabeled trajectories; **UDS** (Yu et al., 2022), which keeps the rewards in the expert demonstrations and simply assigns minimum rewards to the unlabeled data; **OTR** (Luo et al., 2023), which learns a reward function via using the optimal transport to get the optimal alignment between the expert demonstrations and unlabeled trajectories. For a fair comparison, all these methods adopt IQL as the base algorithm. We additionally compare against BC, and 10%BC (Chen et al., 2021). We take the results of IQL+ORIL and IQL+UDS directly from the OTR paper. As OTR computes rewards using pure observations (and SEABO uses $(s, a, s')$ to query the reward), we modify its way of solving optimal coupling by involving actions, and run IQL+OTR on these datasets with its official codebase. We summarize the comparison results in Table 2. It can be found that, though methods like ORIL and OTR can lead to competitive or better performance on some of the datasets than IQL trained with raw rewards, SEABO beats them on numerous tasks. Meanwhile, SEABO is the only method that can surpass IQL with ground-truth rewards in terms of the total score.

**SEABO evaluation on wider datasets.** We further evaluate IQL+SEABO on two challenging domains from D4RL, `AntMaze` and `Adroit`. We run IQL with ground-truth rewards to obtain the IQL performance. We take the results of IQL+OTR from its paper directly. Table 3 demonstrates the detailed comparison results. We find that IQL+SEABO beats IQL and IQL+OTR on 5 out of 6 datasets on `AntMaze`, and outperforms baselines on 6 out of 8 datasets on `Adroit`, often by a large margin. IQL+SEABO incurs a performance improvement of **6.0%** and **32.0%** beyond IQL with vanilla rewards on `AntMaze` and `Adroit` tasks, respectively. These indicate that SEABO with one single expert trajectory can handle datasets with diverse behavior, and work as a good and promising proxy to the hand-crafted rewards.

Table 3: **Experimental results on the AntMaze-v0 and Adroit-v0 domains.** SEABO and OTR use IQL as the base algorithm. IQL denotes that IQL uses the ground-truth reward for policy learning. We report the mean normalized scores and the corresponding standard deviations. We **bold** and highlight the best mean score cell.

| Task Name | IQL | IQL+OTR | IQL+SEABO |
|---|---|---|---|
| umaze | 87.5±2.6 | 83.4±3.3 | **90.0**±1.8 |
| umaze-diverse | 62.2±13.8 | **68.9**±13.6 | 66.2±7.2 |
| medium-diverse | 70.0±10.9 | 70.4±4.8 | **72.2**±4.1 |
| medium-play | 71.2±7.3 | 70.5±6.6 | **71.6**±5.4 |
| large-diverse | 47.5±9.5 | 45.5±6.2 | **50.0**±6.8 |
| large-play | 39.6±5.8 | 45.3±6.9 | **50.8**±8.7 |
| Total Score | 378.0 | 384.0 | **400.8** |

| Task Name | IQL | IQL+OTR | IQL+SEABO |
|---|---|---|---|
| pen-human | 70.7±8.6 | 66.8±21.2 | **94.3**±12.0 |
| pen-cloned | 37.2±7.3 | 46.9±20.9 | **48.7**±15.3 |
| door-human | 3.3±1.3 | **5.9**±2.7 | 5.1±2.0 |
| door-cloned | **1.6**±0.5 | 0.0±0.0 | 0.4±0.8 |
| relocate-human | 0.1±0.0 | 0.1±0.1 | **0.4**±0.5 |
| relocate-cloned | **-0.2**±0.0 | **-0.2**±0.0 | **-0.2**±0.0 |
| hammer-human | 1.6±0.6 | 1.8±1.4 | **2.7**±1.8 |
| hammer-cloned | 2.1±1.0 | 0.9±0.3 | **2.2**±0.8 |
| Total Score | 116.4 | 122.2 | **153.6** |

Table 4: **Comparison of SEABO against imitation learning algorithms**. We use IQL as the base algorithm for SEABO and PWIL. PWIL-action means that we concatenate state and action to compute rewards in PWIL. We report the mean performance at the final 10 episodes of evaluation for each algorithm, ± captures the standard deviation. We highlight the best mean score cell.

| Task Name | SQIL | DemoDICE | SMODICE | PWIL-action | SEABO |
|---|---|---|---|---|---|
| halfcheetah-medium | 31.3±1.8 | 42.5±1.7 | 41.7±1.0 | 44.4±0.2 | **44.8**±0.3 |
| hopper-medium | 44.7±20.1 | 55.1±3.3 | 56.3±2.3 | 60.4±1.8 | **80.9**±3.2 |
| walker2d-medium | 59.6±7.5 | 73.4±2.6 | 13.3±9.2 | 72.6±6.3 | **80.9**±0.6 |
| halfcheetah-medium-replay | 29.3±2.2 | 38.1±2.7 | 38.7±2.4 | **42.6**±0.5 | 42.3±0.1 |
| hopper-medium-replay | 45.2±23.1 | 39.0±15.4 | 44.3±19.7 | **94.0**±7.0 | 92.7±2.9 |
| walker2d-medium-replay | 36.3±13.2 | 52.2±13.1 | 44.6±23.4 | 41.9±6.0 | **74.0**±2.7 |
| halfcheetah-medium-expert | 40.1±6.4 | 85.8±5.7 | 87.9±5.8 | **89.5**±3.6 | 89.3±2.5 |
| hopper-medium-expert | 49.8±5.8 | 92.3±14.2 | 76.0±8.6 | 70.9±35.1 | **97.5**±5.8 |
| walker2d-medium-expert | 35.9±22.2 | 106.9±1.9 | 47.8±31.1 | 109.8±0.2 | **110.9**±0.2 |
| Total Score | 372.2 | 585.3 | 450.6 | 626.1 | **713.3** |

## 5.2 COMPARISON AGAINST OFFLINE IL ALGORITHMS

To further show the advantages of SEABO, we additionally compare it against recent strong offline imitation learning approaches, including DemoDICE (Kim et al., 2022b) and SMODICE (Ma et al., 2022). We also convert two online IL algorithms into the offline setting, SQIL (Reddy et al., 2020) and PWIL (Dadashi et al., 2021), where we replace the base algorithm in SQIL with TD3_BC and utilize IQL as the base algorithm for PWIL. All algorithms are run using their official implementations under the identical experimental setting as SEABO (*i.e.*, one single expert demonstration). For a fair comparison, we involve actions when training discriminators in SMODICE and measuring the distance in PWIL. We use IQL as the base algorithm for SEABO. The empirical results in Table 4 show that IQL+SEABO achieves the best performance on 6 out of 9 datasets, and has the highest total score (surpassing the second highest one by **13.9%**). Though SEABO underperforms PWIL on some datasets, it significantly beats PWIL on tasks like `hopper-medium-v2`. Note that SMODICE behaves poorly on many tasks, which is also observed in Li et al. (2023).

## 5.3 STATE-ONLY REGIMES

We now examine how SEABO behaves when the expert demonstrations consist of only observations, *i.e.*, $\mathcal{D}_e = \{\tau_e^i\}_{i=1}^M$, where $M$ is the size of the demonstration and $\tau = \{s_0, s_1, \ldots, s_T\}$. In principle, SEABO can also calculate rewards by querying the KD-tree with only states, $(\tilde{s}_e, \tilde{s}_e') = $ `NearestNeighbor`$(\mathcal{D}_e, (s, s'))$. The distance can then be calculated with some distance metric $D, d = D((\tilde{s}_e, \tilde{s}_e'), (s, s'))$, and the rewards can be computed accordingly, via Equation 3. For baselines, since DemoDICE and ValueDICE are inapplicable to state-only regimes (Zhu et al., 2021), we compare against LobsDICE (Kim et al., 2022a), which is a state-of-the-art offline IL algorithm that learns from expert observations. We also involve SMODICE, PWIL, and OTR for comparison, and train them using only expert observations. All baselines are run with their official implementa-

Table 5: **Experimental results on the state-only regime**. SEABO, PWIL, and OTR utilize IQL as the base offline RL algorithm. PWIL-state denotes that PWIL only uses observations to compute rewards. The results are averaged over the final 10 evaluations, and $\pm$ captures the standard deviation. We highlight the cell with the best mean performance.

| Task Name | SMODICE | LobsDICE | PWIL-state | OTR | SEABO |
|---|---|---|---|---|---|
| halfcheetah-medium | 41.1±2.1 | 41.5±1.8 | 0.1±0.6 | 43.3±0.2 | **45.0**±0.2 |
| hopper-medium | 56.5±1.8 | 56.9±1.4 | 1.4±0.5 | **78.7**±5.5 | 74.7±5.2 |
| walker2d-medium | 15.5±18.6 | 69.3±5.4 | 0.2±0.2 | 79.4±1.4 | **81.3**±1.3 |
| halfcheetah-medium-replay | 39.2±3.1 | 39.9±3.1 | -2.4±0.2 | 41.3±0.6 | **42.4**±0.6 |
| hopper-medium-replay | 55.3±21.4 | 41.6±16.8 | 0.7±0.2 | 84.8±2.6 | **88.0**±0.7 |
| walker2d-medium-replay | 37.8±10.2 | 33.2±7.0 | -0.2±0.2 | 66.0±6.7 | **76.4**±3.0 |
| halfcheetah-medium-expert | 88.0±4.0 | 89.4±3.2 | 0.0±1.0 | 89.6±3.0 | **91.8**±1.5 |
| hopper-medium-expert | 75.1±11.7 | 53.4±3.2 | 2.7±2.1 | 93.2±20.6 | **97.5**±6.4 |
| walker2d-medium-expert | 32.3±14.7 | 106.6±2.7 | 0.2±0.3 | 109.3±0.8 | **110.5**±0.3 |
| Total Score | 440.8 | 531.8 | 2.7 | 685.6 | **707.6** |

Table 6: **Comparison of different choices of search algorithms in SEABO.** We report the mean normalized scores with standard deviations. We highlight the best mean score cell except for IQL.

| Task Name | IQL | SEABO (KD-tree) | SEABO (Ball-tree) | SEABO (HNSW) |
|---|---|---|---|---|
| halfcheetah-medium | 47.4±0.2 | 44.8±0.3 | **44.9**±0.3 | 42.1±0.6 |
| hopper-medium | 66.2±5.7 | **80.9**±3.2 | 80.7±3.7 | 47.2±2.9 |
| walker2d-medium | 78.3±8.7 | **80.9**±0.6 | 80.8±0.6 | 30.7±19.9 |
| halfcheetah-medium-replay | 44.2±1.2 | 42.3±0.1 | **42.5**±0.3 | 26.9±4.2 |
| hopper-medium-replay | 94.7±8.6 | **92.7**±2.9 | 92.1±2.3 | 25.8±7.5 |
| walker2d-medium-replay | 73.8±7.1 | 74.0±2.7 | **74.3**±2.0 | 29.1±10.1 |
| halfcheetah-medium-expert | 86.7±5.3 | **89.3**±2.5 | 89.2±2.4 | 34.5±2.2 |
| hopper-medium-expert | 91.5±14.3 | **97.5**±5.8 | 96.7±6.2 | 41.5±7.7 |
| walker2d-medium-expert | 109.6±1.0 | **110.9**±0.2 | **110.9**±0.1 | 108.6±0.8 |
| Total Score | 692.4 | **713.3** | 712.1 | 386.4 |

tions and single expert demonstration. The results in Table 5 suggest that SEABO outperforms other methods on 8 out of 9 tasks, achieving a total score of **707.6**, while LobsDICE and OTR only have a total score of 531.8 and 685.6, respectively. It indicates that SEABO can work quite well regardless of whether the expert demonstrations contain actions, further demonstrating the advantages of SEABO. Note that the failure of PWIL in state-only regimes is also reported in Luo et al. (2023).

## 5.4 COMPARISON OF DIFFERENT SEARCH ALGORITHMS

The most critical component in SEABO is the nearest neighbor search algorithm. It is interesting to check how SEABO performs under different search algorithms. To that end, we build SEABO on top of Ball-tree (Omohundro, 1989; Liu et al., 2006), and HNSW (Hierarchical Navigable Small World graphs, Malkov & Yashunin (2018)). These are widely applied nearest neighbor algorithms, where Ball-tree partitions regions via hyper-spheres and HNSW is a fully graph-based search structure. We allow the single expert demonstration to involve actions (*i.e.*, query with $(s, a, s')$), and run all of the variants of SEABO using the same set of hyperparameters for a fair comparison. Empirical results on 9 D4RL locomotion datasets are shown in Table 6. It is interesting to see that SEABO with Ball-tree is competitive with SEABO with KD-tree (their performance differences are minor), while SEABO with HNSW exhibits poor performance on many datasets. This means that the choice of the search algorithm counts in SEABO, and simply employing KD-tree can already guarantee good performance. Please see more discussions in Appendix C.

## 5.5 PARAMETER STUDY

It is vital to examine how sensitive SEABO is to the introduced hyperparameters. Due to the space limit, we can only report part of the experiments here and defer more experiments to Appendix B.3.

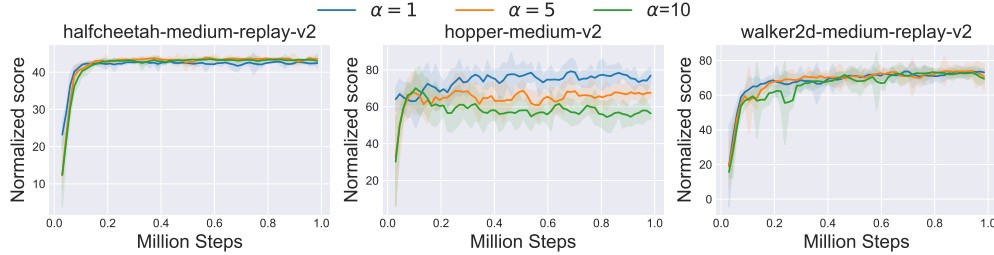

Figure 3: **Parameter study on the reward scale.** The shaded region denotes the standard deviation.

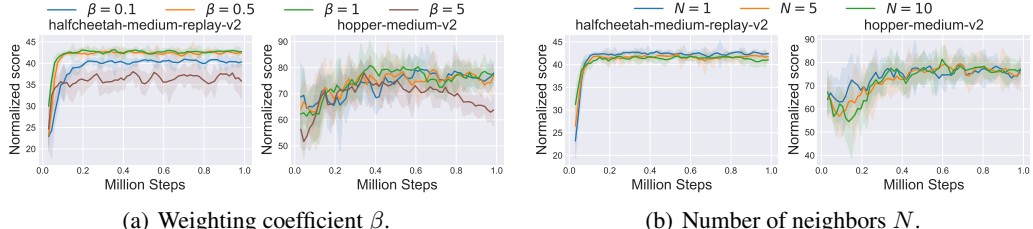

(a) Weighting coefficient $\beta$.        (b) Number of neighbors $N$.

Figure 4: **Parameter study of (a) weighting coefficient $\beta$, (b) number of neighbors $N$.** The shaded region captures the standard deviation.

**Reward scale $\alpha$.** $\alpha$ controls the scale of the resulting rewards. To check its influence, we conduct experiments on three datasets from D4RL locomotion tasks and sweep $\alpha$ across $\{1, 5, 10\}$. Results in Figure 3 demonstrate that the best $\alpha$ may depend on the dataset while a smaller $\alpha$ is preferred.

**Weighting coefficient $\beta$.** $\beta$ is probably the most critical hyperparameter which decides the scale of the distance. In Figure 4(a), we vary $\beta$ across $\{0.1, 0.5, 1, 5\}$, and find that the performance drops with too small or too large $\beta$. It seems that $\beta = 0.5$ or $\beta = 1$ can achieve a good trade-off.

**Number of neighbors $N$.** To see whether the number of neighbors $N$ matters, we run IQL+SEABO with $N \in \{1, 5, 10\}$. Results in Figure 4(b) show that SEABO is robust to this hyperparameter.

**Number of expert demonstrations $K$.** We investigate whether increasing the number of expert demonstrations can further boost the performance of SEABO and baselines by running experiments of these methods on 9 MuJoCo locomotion tasks. We report the aggregate performance (*i.e.*, total score) in Table 7. One can see that all methods enjoy performance improvement when $K = 10$, while none of them can outperform SEABO (there still exists a large performance gap).

Table 7: **Comparison of SEABO against baseline algorithms under different amounts of expert demonstrations**. We report the aggregate performances and **bold** the best one.

| # demo | DemoDICE | IQL+ORIL | IQL+UDS | IQL+OTR | IQL+PWIL | IQL+SEABO |
|--------|----------|----------|---------|---------|----------|-----------|
| $K = 1$ | 585.3 | 588.5 | 495.1 | 685.6 | 626.1 | **713.3** |
| $K = 10$ | 589.3 | 618.3 | 575.8 | 694.2 | 638.0 | **716.1** |

## 6   CONCLUSION

In this paper, we propose a novel search-based offline imitation learning method, dubbed SEABO, that annotates the unlabeled offline trajectories in an unsupervised learning manner. SEABO builds a KD-tree using the expert demonstration(s), and searches the nearest neighbors of the query sample. We then measure their distance and output the reward signal via a squashing function. SEABO is easy to implement and can be incorporated with any offline RL algorithm. Experiments on D4RL datasets show that SEABO can incur competitive or even better offline policies than pre-defined reward functions. SEABO can also function well if the expert demonstrations are made up of only observations. For future work, it is interesting to apply SEABO in visual offline RL datasets (*e.g.*, Lu et al. (2022b)), or adapt SEABO to cross-domain offline imitation learning tasks.

ACKNOWLEDGEMENTS

This work was supported by the STI 2030-Major Projects under Grant 2021ZD0201404 and NSFC under Grant 62250068. The authors would like to thank the anonymous reviewers for their valuable comments and advice.

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

# A   HYPERPARAMETER SETUP

In this section, we detail the hyperparameter setup utilized in our experiments. We conduct experiments on 9 MuJoCo locomotion "-v2" medium-level datasets, 6 AntMaze "-v0" datasets, and 8 Adroit "-v0" datasets, yielding a total of **23** tasks. We list the hyperparameter setup for IQL and TD3_BC on MuJoCo locomotion tasks in Table 8. We keep the hyperparameter setup of the base offline RL algorithms unchanged for both IQL and TD3_BC. For IQL, we do not rescale the rewards in the datasets by $^{1000}/_{\text{max\_return}-\text{min\_return}}$, as we have an additional hyperparameter $\alpha$ to control the reward scale. In practice, we find minor performance differences if we rescale the rewards. We generally utilize the same formula of squashing function for most of the datasets, except that we set $\beta = 1$ in hopper-medium-replay-v2, and $\alpha = 10, \beta = 0.1$ in hopper-medium-expert-v2 for better performance. Note that using $\alpha = 1, \beta = 0.5$ on these tasks can also produce a good performance (*e.g.*, setting $\alpha = 1, \beta = 0.5$ on hopper-medium-replay-v2 leads to an average performance of 87.2, still outperforming strong baselines like OTR), while slightly modifying the hyperparameter setup can result in better performance. We divide the scaled distance by the action dimension of the task to strike a balance between different tasks (as we use one set of hyperparameters). This is also adopted in PWIL paper (Dadashi et al., 2021). For TD3_BC, we use the same type of squashing function as IQL on the locomotion tasks, with $\alpha = 1, \beta = 0.5$, except that we use $\alpha = 10$ for walker2d-medium-v2 and walker2d-medium-replay-v2 for slightly better performance. We use the official implementation of TD3_BC (https://github.com/sfujim/TD3_BC) and adopt the PyTorch (Paszke et al. (2019)) version of IQL for evaluation.

Table 8: Hyperparameter setup of SEABO on locomotion tasks, with IQL and TD3_BC as the base offline RL algorithms.

|  | Hyperparameter | Value |
|---|---|---|
| Shared Configurations | Hidden layers | $(256, 256)$ |
|  | Discount factor | 0.99 |
|  | Actor learning rate | $3 \times 10^{-4}$ |
|  | Critic learning rate | $3 \times 10^{-4}$ |
|  | Batch size | 256 |
|  | Optimizer | Adam (Kingma & Ba, 2015) |
|  | Target update rate | $5 \times 10^{-3}$ |
|  | Activation function | `ReLU` |
| IQL | Value learning rate | $3 \times 10^{-4}$ |
|  | Temperature | 3.0 |
|  | Expectile | 0.7 |
| TD3_BC | Policy noise | 0.2 |
|  | Policy noise clipping | $(-0.5, 0.5)$ |
|  | Policy update frequency | 2 |
|  | Normalization weight | 2.5 |
| SEABO | Squashing function | $r = \exp(-\frac{0.5 \times d}{|\mathcal{A}|})$ |
|  | Distance measurement | `Euclidean distance` |
|  | Number of neighbors | 1 |
|  | Number of expert demonstrations | 1 |

We summarize the hyperparameter setup of SEABO (using IQL as the underlying algorithm) on the AntMaze domain and Adroit domain in Table 9 and Table 10, respectively. We only list the different hyperparameters in these tables and the other hyperparameters follow those presented in Table 8. Note that we filter the highest return trajectory as the expert demonstration in the Adroit domain, while selecting the goal-reached trajectory as the expert demonstration in the AntMaze domain, which is also adopted in OTR paper (Luo et al., 2023). We adopt a comparatively large $\beta = 5$ on AntMaze tasks. We also follow the IQL paper (Kostrikov et al., 2022) to subtract 1 from the rewards, which we find can result in better performance. For Adroit tasks, we remove the action dimension in the squashing function, since these tasks have large action space dimensions. If one insists on involving $|\mathcal{A}|$, a much larger $\beta$ than 0.5 is then necessary to mitigate its influence. We find

Table 9: Hyperparameter setup of SEABO on AntMaze tasks, with IQL as the base offline RL algorithm.

| | Hyperparameter | Value |
|---|---|---|
| IQL | Temperature | 10.0 |
| | Expectile | 0.9 |
| SEABO | Squashing function | $r = \exp(-\frac{5 \times d}{|\mathcal{A}|}) - 1$ |

Table 10: Hyperparameter setup of SEABO on Adroit tasks, with IQL as the base offline RL algorithm.

| | Hyperparameter | Value |
|---|---|---|
| IQL | Temperature | 0.5 |
| | Expectile | 0.7 |
| | Actor dropout rate | 0.1 |
| SEABO | Squashing function | $r = \exp(-0.5 \times d)$ |

that simply removing $|\mathcal{A}|$ can ensure quite good performance on all of the evaluated Adroit datasets. Note that OTR (Luo et al., 2023) also adopts different forms of squashing functions for different domains. We query with $(s, s')$ for Adroit tasks and $(s, a, s')$ for other domains.

To acquire expert demonstrations, we use the trajectory with the highest return as expert demonstrations on MuJoCo locomotion tasks and Adroit tasks, and filter the goal-reached trajectory in AntMaze tasks. For all of the baseline reward learning and offline imitation learning algorithms, we follow this setting and run them with their official codebases[2] over five different random seeds. We use the PWIL implementation from Acme (Hoffman et al., 2020)[3].

In SEABO, we use the KD-tree implementation from the scipy library (Virtanen et al., 2020), *i.e.*, `scipy.spatial.KDTree`. We set the number of nearest neighbors $N = 1$, and keep other default hyperparameters in KD-tree. Note that we can directly get the desired distance by querying the KD-tree. For Ball-tree, we use its implementation in the scikit-learn package (Pedregosa et al., 2011), *i.e.*, `sklearn.neighbors.BallTree`. We also keep its original hyperparameters unchanged. For HNSW, we use its implementation in `hnswlib`[4]. We use the suggested hyperparameter setting in its GitHub page and set `ef_construction=200` (which defines a construction time/accuracy trade-off) and `M=16` (which defines the maximum number of outgoing connections in the graph). All these search algorithms adopt the Euclidean distance as the distance measurement.

In our experiments, we use `MuJoCo 2.0` (Todorov et al., 2012) with `Gym` version `0.18.3`, `PyTorch` (Paszke et al., 2019) version `1.8`. We use the *normalized score* metric recommended in the D4RL paper (Fu et al., 2020), where 0 corresponds to a random policy, and 100 corresponds to an expert policy. Formally, suppose we get the average return $J$ by deploying the learned policy in the test environment, the normalized score gives:

$$\text{Normalized score} = \frac{J - J_r}{J_e - J_r} \times 100, \tag{4}$$

where $J_r$ is the return of a random policy, and $J_e$ is the return of an expert policy.

## B  MISSING EXPERIMENTAL RESULTS

In this section, we present the missing experimental results from the main text due to the space limit.

---

[2]OTR official codebase: https://github.com/ethanluoyc/optimal_transport_reward/. SMODICE official codebase: https://github.com/JasonMa2016/SMODICE. DemoDICE and LobsDICE official codebase: https://github.com/geon-hyeong/imitation-dice.

[3]https://github.com/deepmind/acme/tree/master/acme/agents/jax/pwil

[4]https://github.com/nmslib/hnswlib

Table 11: **Comparison of SEABO against baseline algorithms under 10 expert demonstrations**. We use IQL as the base algorithm for SEABO, PWIL, and OTR. We report the mean performance at the final 10 evaluations for each algorithm, and $\pm$ captures the standard deviation.

| Task Name | IQL | PWIL-state | PWIL-action | OTR-state | OTR-action | SEABO |
|---|---|---|---|---|---|---|
| halfcheetah-medium | 47.4±0.2 | 1.6±1.2 | **47.5±0.2** | 43.1±0.3 | 43.4±0.3 | 44.4±0.2 |
| hopper-medium | 66.2±5.7 | 2.1±1.3 | 70.4±4.2 | 80.0±5.2 | 75.4±4.6 | **81.4±3.5** |
| walker2d-medium | 78.3±8.7 | 0.9±1.3 | **81.9±1.0** | 79.2±1.3 | 79.7±1.2 | 81.1±0.7 |
| halfcheetah-medium-replay | 44.2±1.2 | -2.3±0.5 | **44.6±1.1** | 41.6±0.3 | 41.9±0.3 | 43.9±0.2 |
| hopper-medium-replay | 94.7±8.6 | 1.4±1.2 | **89.7±4.9** | 84.4±1.8 | 85.3±1.1 | 86.4±1.4 |
| walker2d-medium-replay | 73.8±7.1 | -0.1±0.2 | 72.2±10.6 | 71.8±3.8 | 69.1±4.6 | **78.0±0.7** |
| halfcheetah-medium-expert | 86.7±5.3 | -0.3±1.5 | 88.6±4.3 | 87.9±3.4 | 88.3±5.1 | **90.5±2.5** |
| hopper-medium-expert | 91.5±14.3 | 1.5±0.6 | 32.9±25.0 | 96.6±21.5 | 86.6±22.9 | **100.0±7.0** |
| walker2d-medium-expert | 109.6±1.0 | 1.0±1.9 | 110.2±0.2 | 109.6±0.5 | 109.2±0.5 | **110.4±0.6** |
| Total Score | 692.4 | 5.8 | 638.0 | 694.2 | 678.9 | **716.1** |

## B.1 NUMERICAL COMPARISON UNDER TEN EXPERT DEMONSTRATIONS

In Section 5.5, we present the comparison results of SEABO and baseline reward learning and offline IL algorithms under different numbers of expert demonstrations $K \in \{1, 10\}$. However, we only report the aggregate performance (*i.e.*, the total score) on the 9 MuJoCo locomotion medium-level tasks (`medium`, `medium-replay`, `medium-expert`) in Table 7. To make the comparison clearer, we present the detailed normalized scores of these methods under $K = 10$ on different datasets in Table 11, where we mainly compare SEABO against different variants of PWIL and OTR. SEABO computes the rewards with actions involved in the single expert demonstration here.

The results reveal that SEABO outperforms baseline methods on 5 out of 9 datasets and is competitive with baselines on the rest of the datasets. SEABO achieves a total score of **716.1**, surpassing the second best method (OTR-state) by **3.2%**. Though we observe that PWIL-action beats SEABO on datasets like `halfcheetah-medium-v2`, it can perform poorly on datasets like `hopper-medium-expert-v2`. We also note that the performance of PWIL deteriorates in the state-only regimes, *i.e.*, learning from pure expert observations. This phenomenon is also reported in Dadashi et al. (2021); Luo et al. (2023). SEABO, instead, is flexible and can be applied regardless of whether the expert demonstrations contain actions.

Furthermore, we show in Table 12 the results of IQL+SEABO on AntMaze and Adroit datasets when 10 expert demonstrations are provided. We compare IQL+SEABO against IQL+OTR and IQL with raw rewards (denoted as IQL). The results demonstrate that SEABO can recover the performance of the offline RL algorithm with ground-truth rewards and sometimes yield better performance. This advantage is agnostic to the number of expert demonstrations $K$.

IQL+SEABO matches the performance of IQL+OTR on many AntMaze tasks and outperforms IQL+OTR on 6 out of 8 datasets from the Adroit domain. On both the AntMaze domain and Adroit domain, OTR underperforms SEABO in terms of the total score. One may notice that the performance of IQL+SEABO decreases with more expert demonstrations, mainly on the Adroit datasets. This is caused by the performance drop on `pen-human-v0`, which dominate the total score (the magnitude of its score is much larger than those of other datasets). One can also observe that the performance of IQL+OTR declines on many Adroit tasks, given 10 expert demonstrations (see Table 7 in Luo et al. (2023)). Still, IQL+SEABO exhibits strong performance across numerous datasets.

## B.2 COMPARISON OF TD3_BC+OTR AND TD3_BC+SEABO

Since the majority of the experiments in the main text are conducted using IQL as the base offline RL algorithm, it is interesting to see how SEABO competes against baseline methods with another offline RL algorithm as the base method. To that end, we choose TD3_BC and incorporate it with the strong baseline method, OTR. We follow the experimental setting utilized in the main text, filter a single expert demonstration with the highest return in the offline dataset, and deem it as the expert demonstration. We run TD3_BC+SEABO and TD3_BC+OTR on 9 D4RL MuJoCo locomotion datasets. We follow our experimental setup specified in Appendix A, and use the default hyperpa-

Table 12: **Experimental results of SEABO on the AntMaze-v0 and Adroit-v0 domains with 10 expert demonstrations.** SEABO and OTR use IQL as the base algorithm. The average normalized scores along with the corresponding standard deviations are reported. We **bold** and highlight the best mean score cell.

| Task Name | IQL | IQL+OTR | IQL+SEABO |
|---|---|---|---|
| umaze | 87.5±2.6 | **88.7±3.5** | 87.6±2.0 |
| umaze-diverse | 62.2±13.8 | 64.4±18.2 | **70.0±9.5** |
| medium-diverse | 70.0±10.9 | **70.5±6.9** | 70.2±5.4 |
| medium-play | 71.2±7.3 | 72.7±6.2 | **72.8±1.6** |
| large-diverse | 47.5±9.5 | **50.7±6.9** | 50.0±7.9 |
| large-play | 39.6±5.8 | **51.2±7.1** | 48.6±9.8 |
| Total Score | 378.0 | 398.2 | **399.2** |

| Task Name | IQL | IQL+OTR | IQL+SEABO |
|---|---|---|---|
| pen-human | 70.7±8.6 | 69.4±21.5 | **85.8±16.1** |
| pen-cloned | 37.2±7.3 | 42.7±25.0 | **49.2±12.2** |
| door-human | 3.3±1.3 | 4.2±2.1 | **6.8±5.6** |
| door-cloned | **1.6±0.5** | 0.0±0.0 | 0.1±0.1 |
| relocate-human | **0.1±0.0** | **0.1±0.1** | **0.1±0.1** |
| relocate-cloned | **-0.2±0.0** | **-0.2±0.0** | **-0.2±0.0** |
| hammer-human | 1.6±0.6 | 1.4±0.2 | **1.7±0.3** |
| hammer-cloned | **2.1±1.0** | 1.3±0.7 | 1.7±0.5 |
| Total Score | 116.4 | 118.9 | **145.2** |

Table 13: **Comparison of SEABO against OTR using TD3_BC as the base algorithm.** We report the average normalized scores and their standard deviations. We bold and highlight the mean score cell except for TD3_BC. We adopt one single expert demonstration for OTR and SEABO.

| Task Name | BC | 10%BC | TD3_BC | TD3_BC+OTR | TD3_BC+SEABO |
|---|---|---|---|---|---|
| halfcheetah-medium | 42.6 | 42.5 | 48.0±0.7 | 42.6±1.0 | **45.9±0.3** |
| hopper-medium | 52.9 | 56.9 | 60.7±12.5 | 66.4±10.3 | **76.1±4.2** |
| walker2d-medium | 75.3 | 75.0 | 83.7±5.3 | **76.9±5.4** | 76.6±0.4 |
| halfcheetah-medium-replay | 36.6 | 40.6 | 44.4±0.8 | 39.4±1.3 | **43.0±0.4** |
| hopper-medium-replay | 18.1 | 75.9 | 64.8±25.5 | 74.9±28.8 | **96.3±3.0** |
| walker2d-medium-replay | 26.0 | 62.5 | 87.4±8.4 | 69.7±16.4 | **73.1±2.2** |
| halfcheetah-medium-expert | 55.2 | 92.9 | 93.5±2.0 | 74.8±20.1 | **95.7±0.4** |
| hopper-medium-expert | 52.5 | 110.9 | 100.2±20.0 | 103.2±13.9 | **107.1±3.3** |
| walker2d-medium-expert | 107.5 | 109.0 | 109.5±0.5 | 109.0±0.6 | **109.7±0.2** |
| Total Score | 466.7 | 666.2 | 692.3 | 656.9 | **723.5** |

rameter setup of OTR suggested by the authors. We summarize the comparison results in Table 13. It turns out that TD3_BC+SEABO outperforms TD3_BC+OTR on 8 out of 9 datasets, often by a large margin, surpassing it by **10.1%** in terms of the total score. TD3_BC+SEABO is the only algorithm that even beats TD3_BC learned with raw rewards in total score. We observe that the standard deviation of TD3_BC+OTR is large on datasets like `halfcheetah-medium-expert-v2`, while the standard deviation of TD3_BC+SEABO is much smaller. This evidence indicates that SEABO is superior to OTR when acting as the reward labeler, and can consistently aid different base offline RL algorithms recover its performance under ground-truth rewards or achieve better performance.

## B.3 HYPERPARAMETER SENSITIVITY

In Section 5.5, we are only able to attach the results on a small proportion of datasets from D4RL, *e.g.*, `halfcheetah-medium-replay-v2` due to the space limit. In this part, we include wider experimental results in terms of the reward scale $\alpha$, weighting coefficient $\beta$, and number of neighbors $N$. Again, we use IQL as the base offline RL algorithm for SEABO. The expert demonstrations utilized here contain actions. We follow the hyperparameter setup specified in Section A.

**Reward scale $\alpha$.** The reward scale $\alpha$ controls the magnitude of the computed rewards. In Figure 3 of the main text, we find that a smaller $\alpha$ seems to be better (especially on `hopper-medium-v2`). We further conduct experiments on three additional tasks, `halfcheetah-medium-expert-v2`, `hopper-medium-replay-v2`, and `walker2d-medium-v2` by varying $\alpha \in \{1, 5, 10\}$. The results are shown in Figure 5, where we actually do not find much performance difference of $\alpha$ on these three tasks. That indicates that IQL+SEABO is robust to $\alpha$ on most of the datasets. In practice, one can simply set $\alpha = 1$, which we find can already yield very good performance on MuJoCo tasks, AntMaze tasks, and Adroit tasks.

**Weighting coefficient $\beta$.** As commented in the main text, the weighting coefficient $\beta$ is perhaps the most important hyperparameter in SEABO, since it controls the weights of the measured distance and this may have a significant influence on the final rewards. For a specific domain,

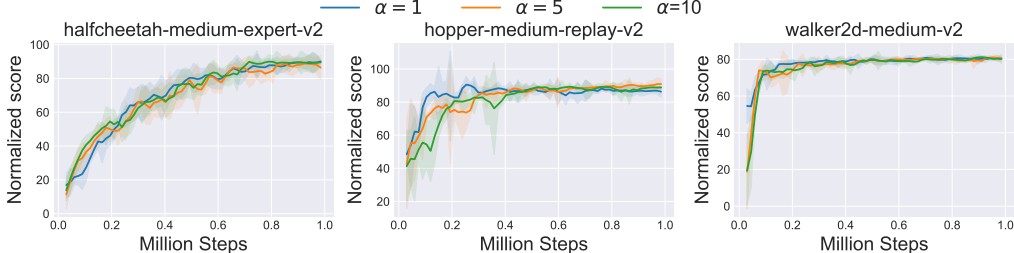

Figure 5: **Additional experiments on the influence of** $\alpha$**.** The shaded region captures the standard deviation. All other hyperparameters are kept unchanged except $\alpha$.

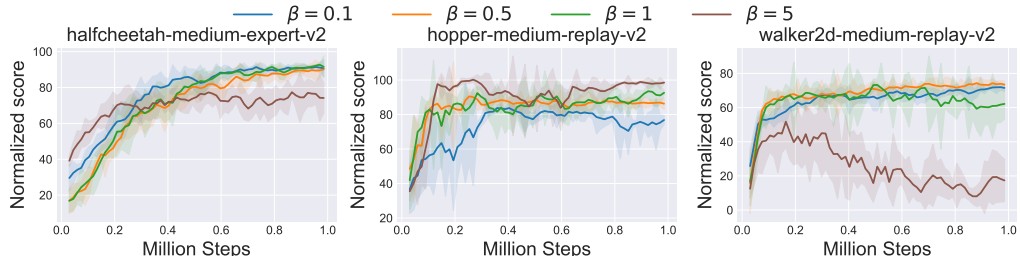

Figure 6: **Additional experiments on the effect of** $\beta$. We choose three additional datasets from D4RL, and plot their mean normalized score curve. The shaded area denotes the standard deviation.

we mostly adopt a fixed $\beta$ as we do not want to bother tuning this hyperparameter. However, we believe it is vital to examine how $\beta$ influences the performance of SEABO in wider experiments. We additionally conduct several experiments on `halfcheetah-medium-expert-v2`, `hopper-medium-replay-v2`, `walker2d-medium-replay-v2` from D4RL locomotion tasks. We sweep $\beta$ across $\{0.1, 0.5, 1, 5\}$, and summarize the results in Figure 6. It can be clearly seen that a large $\beta$ results in poor performance on `halfcheetah-medium-expert-v2` and `walker2d-medium-replay-v2`, while setting $\beta = 5$ results in the best performance on `hopper-medium-replay-v2`. In the hyperparameter setup part, we state that we set $\beta = 1$ on `hopper-medium-replay-v2` due to the fact that SEABO is comparatively stable with $\beta\{0.5, 1\}$. We do not doubt that the best $\beta$ is task-dependent, and one can get higher performance by carefully tuning this hyperparameter. However, we empirically show that using a fixed $\beta$ is also feasible, and we believe this is appealing since the users can get rid of the work of tedious hyperparameter search.

**Number of neighbors** $N$**.** The number of neighbors $N$ is a hyperparameter introduced in the nearest neighbor algorithms. For all of our main experiments, we simply adopt $N = 1$, *i.e.*, searching for the nearest neighbor. In Figure 4(b), we see that SEABO is robust to this hyperparameter. To examine whether this conclusion applies to a wider range of datasets, we conduct experiments on three additional datasets, `halfcheetah-medium-v2`, `halfcheetah-medium-expert-v2`, and `walker2d-medium-replay-v2`. The results are summarized in Figure 7, where we also observe that SEABO is robust to this hyperparameter, indicating the effectiveness and generality of SEABO.

## B.4 PERFORMANCE OF SEABO UNDER LONG-HORIZON MANIPULATION TASKS

In this part, we investigate how SEABO behaves under long-horizon manipulation tasks. To that end, we evaluate SEABO in Kitchen datasets (Fu et al., 2020). The kitchen environment (Gupta et al., 2019) consists of a 9 DoF Franka robot interacting with a kitchen scene that includes an openable microwave, four turnable oven burners, an oven light switch, a freely movable kettle, two hinged cabinets, and a sliding cabinet door. In kitchen, the robot may need to manipulate different components, e.g., it may need to open the microwave, move the kettle, turn on the light, and slide open the cabinet (precision is required). We run IQL+SEABO on three kitchen datasets using the author-recommended hyperparameters of IQL on the kitchen environment. We set reward

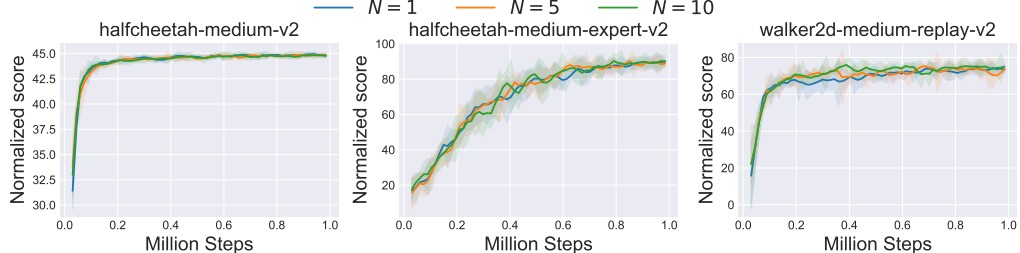

Figure 7: **Additional experiments on examining the influence of the number of neighbors in KD-tree**. The shaded region represents the standard deviation.

Table 14: **Comparison of SEABO against baselines in the Kitchen tasks.** We report the average normalized scores and the corresponding standard deviations. We bold and highlight the best mean score cell.

| Task Name | BC | CQL | IQL | IQL+SEABO |
|---|---|---|---|---|
| kitchen-complete-v0 | 65.0 | 43.8 | 62.5 | **67.5**±4.2 |
| kitchen-partial-v0 | 38.0 | 49.8 | 46.3 | **71.0**±4.1 |
| kitchen-mixed-v0 | 51.5 | 51.0 | 51.0 | **55.0**±3.5 |
| Average Score | 51.5 | 48.2 | 53.3 | **64.5** |

scale $\alpha = 1$, coefficient $\beta = 0.5$ for SEABO. We compare IQL+SEABO against some baselines taken from the IQL paper and summarize the results in Table 14. We find that SEABO exhibits superior performance, surpassing IQL with raw rewards by **21.0%**. We believe these results show that SEABO can aid some long-horizon manipulation tasks.

However, we experimentally find that SEABO does not exhibit strong performance for some tasks that require high precision, e.g., the IKEA Furniture assembly benchmark (Lee et al., 2019; 2021; Heo et al., 2023). We leave the open problem of how to enable SEABO to successfully address such benchmarks a future work.

### B.5 LEARNING CURVES

In this section, we provide the detailed training curves of IQL+SEABO on the locomotion tasks, AntMaze tasks, and Adroit tasks. We also provide learning curves of TD3_BC+SEABO on locomotion tasks. We summarize the results of IQL+SEABO on D4RL MuJoCo locomotion tasks in Figure 8, the performance of IQL+SEABO on AntMaze tasks in Figure 9, and the curves of IQL+SEABO on Adroit tasks in Figure 10. The results of TD3_BC+SEABO are depicted in Figure 11.

From all these results, we find that both IQL+SEABO and TD3_BC+SEABO have stable and strong performance on the evaluated tasks, indicating the advantages of our method.

## C DISCUSSIONS ON DIFFERENT SEARCH ALGORITHMS

The success of SEABO can be largely attributed to the adopted search algorithm (*i.e.*, KD-tree). In Section 5.4 of the main text, we compare different design choices for the underlying search algorithm. It is not surprising to find that Ball-tree results in a similar performance as KD-tree, as Ball-tree shares many similarities with KD-tree. However, we find that HNSW incurs quite poor performance on many datasets using its default hyperparameter setup (see Appendix A). HNSW builds a multi-layer structure made up of a hierarchical set of proximity graphs for nested subsets of the stored elements while employing a heuristic for selecting proximity graph neighbors. HNSW is a graph-based search algorithm. Based on the empirical results in Table 6 in the main text, we find that HNSW leads to quite poor performance for the base offline RL algorithm, only achieving competitive performance against KD-tree on `halfcheetah-medium-v2` and `walker2d-medium-expert-v2`.

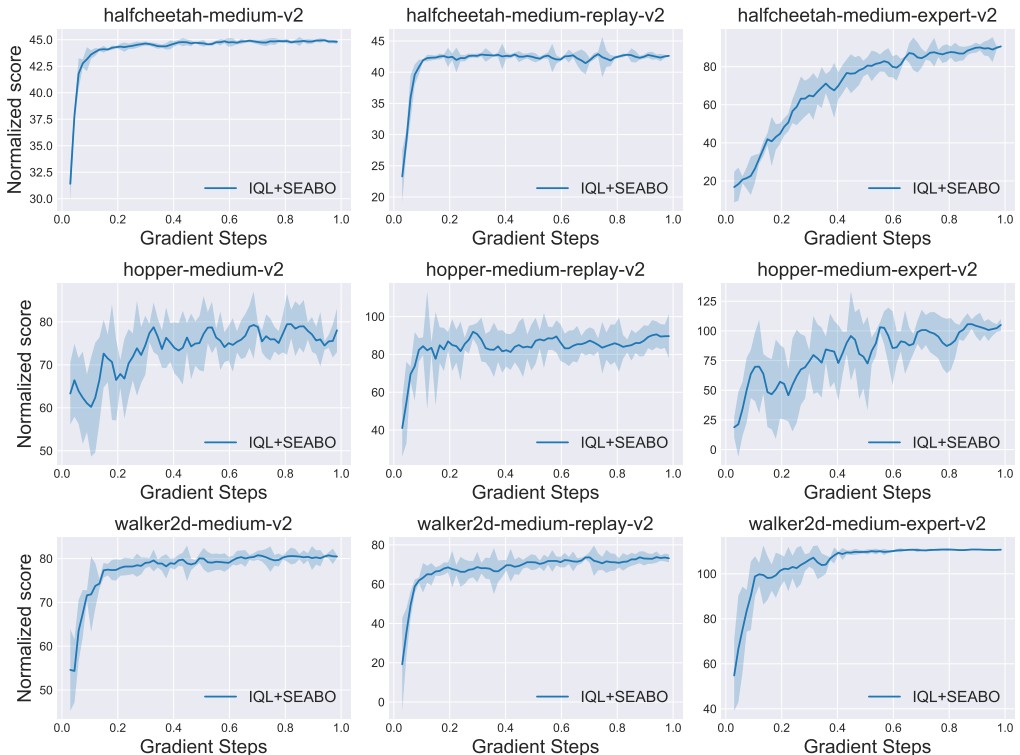

Figure 8: **Full learning curves of IQL+SEABO on D4RL MuJoCo datasets.** We plot the average performance and the shaded region captures the standard deviation.

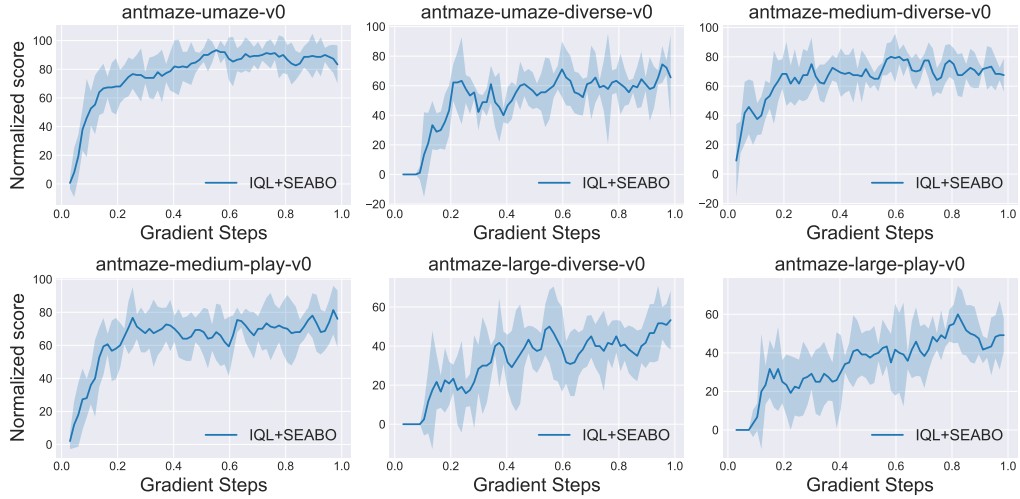

Figure 9: **Full learning curves of IQL+SEABO on AntMaze tasks.** The mean performance in conjunction with the standard deviations are plotted.

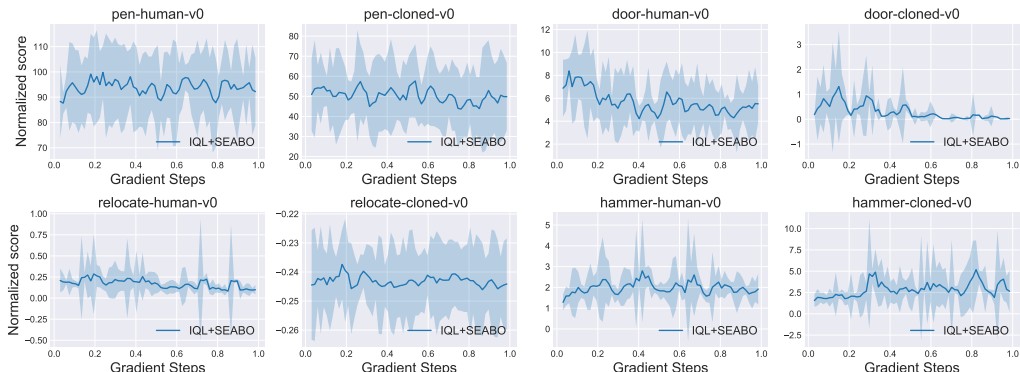

Figure 10: **Full learning curves of IQL+SEABO on Adroit datasets.** We report the mean performance along with its standard deviation.

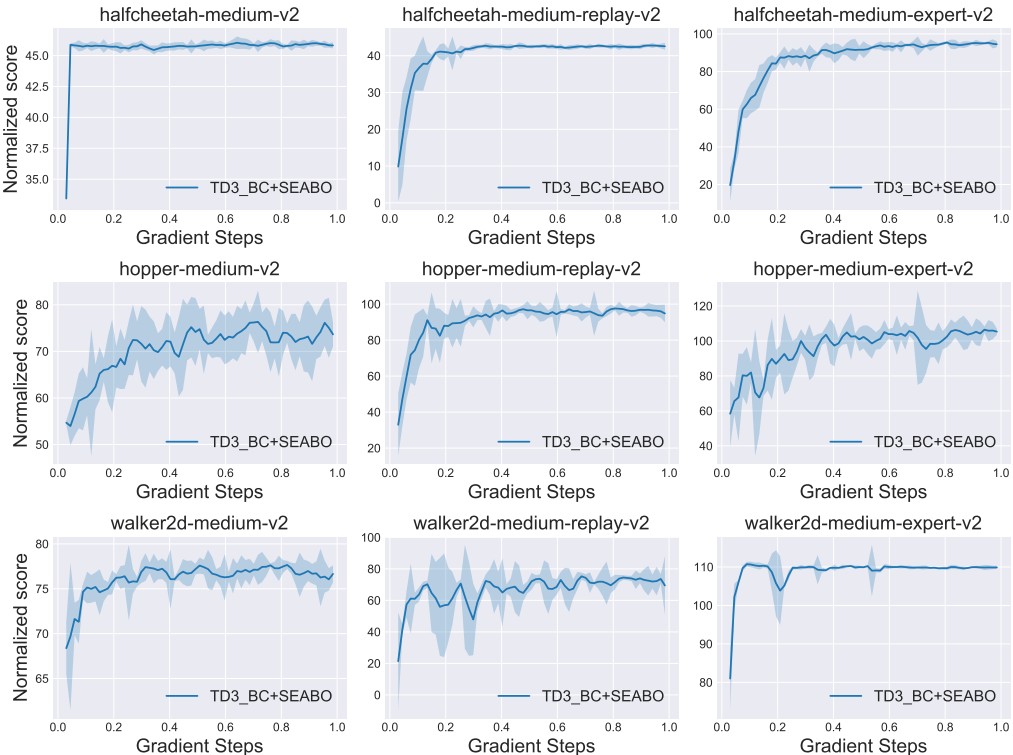

Figure 11: **Full learning curves of TD3_BC+SEABO on D4RL MuJoCo datasets.** The average performance as well as its statistical significance is depicted.

In this subsection, we try to understand why HNSW fails through some empirical evidence. We choose some subsets, `halfcheetah-medium-v2`, `halfcheetah-medium-expert-v2`, `hopper-medium-replay-v2`, `hopper-medium-expert-v2`, `walker2d-medium-v2`, and `walker2d-medium-replay-v2`, from D4RL MuJoCo datasets and plot the reward density of ground-truth rewards, rewards computed using KD-tree, and rewards acquired via HNSW. We summarize the results in Figure 12. It is clear that SEABO with KD-tree can produce a similar reward structure as the ground-truth reward distribution, while SEABO with HNSW tends to assign large rewards to only a small proportion of samples and small rewards to the majority of transitions. We believe this explains the unsatisfying performance of IQL+SEABO with HNSW as the base search algorithm, indicating that a graph-based search mechanism may not be suitable for D4RL datasets. Another possible explanation is that the hyperparameters of HNSW need to be tuned to adapt to different tasks. We do not doubt that a careful tuning of hyperparameters (*e.g.*, the maximum number of outgoing connections in the graph, the number of neighbors, etc.) has the potential of making SEABO with HNSW work in D4RL datasets. However, we do not think it is necessary to do that considering the fact that adopting KD-tree with its default hyperparameters can already result in quite good performance across different datasets. Hence, it is recommended that one uses KD-tree (or Ball-tree) as the base search algorithm.

## D  COMPUTE INFRASTRUCTURE

In Table 15, we list the compute infrastructure that we use to run all of the algorithms.

Table 15: Compute infrastructure.

| CPU | GPU | Memory |
|---|---|---|
| AMD EPYC 7452 | RTX3090×8 | 288GB |

## E  LIMITATIONS

Despite the simplicity and effectiveness of our proposed algorithm, SEABO, we have to admit honestly that there may exist some potential limitations. First, SEABO is slightly sensitive to the weighting coefficient $\beta$ on *some datasets* (not all datasets), and one may need to manually tune it so as to find the best-suited hyperparameter setup for a specific task. While based on our empirical results, one can find the best $\beta \in \{0.5, 1, 5\}$ using grid search, It is not difficult to conduct experiments since SEABO is computationally efficient (and can be applied with only CPUs). Second, it may take more time for SEABO to annotate the unlabeled trajectories with visual input, as images are hard to process. Whereas, we can preprocess the visual images using some pre-trained image encoder (*e.g.*, ImageNet pretrain models) to obtain low-dimensional representations of the high-dimensional image. Note that we build KD-tree upon expert demonstrations which usually contain a small amount of expert transitions. Thus, it should not be time-consuming to annotate the visual trajectories.

We hope this work can provide some new insights to the community and inspire future work on offline imitation learning.

## F  ADDITIONAL REWARD PLOTS ON ADROIT AND ANTMAZE TASKS

In this section, we provide reward distribution plots of the ground truth rewards, rewards obtained by SEABO, and rewards output from HNSW on some Adroit-v0 and AntMaze-v0 tasks, `hammer-human-v0`, `hammer-cloned-v0`, `door-human-v0`, `door-cloned-v0`, `antmaze-uamze-v0`, and `antmaze-medium-diverse-v0`. Note that we provide the histogram plot of rewards in `antmaze-medium-diverse-v0` as most of the samples in this datasets have quite similar reward signals, making it difficult to draw the density plot. We summarize the results in Figure 13. It can be seen that with KD-tree, SEABO outputs similar reward density as vanilla rewards (e.g., SEABO successfully gives three peaks in `hammer-human-v0` and `door-human-v0`).

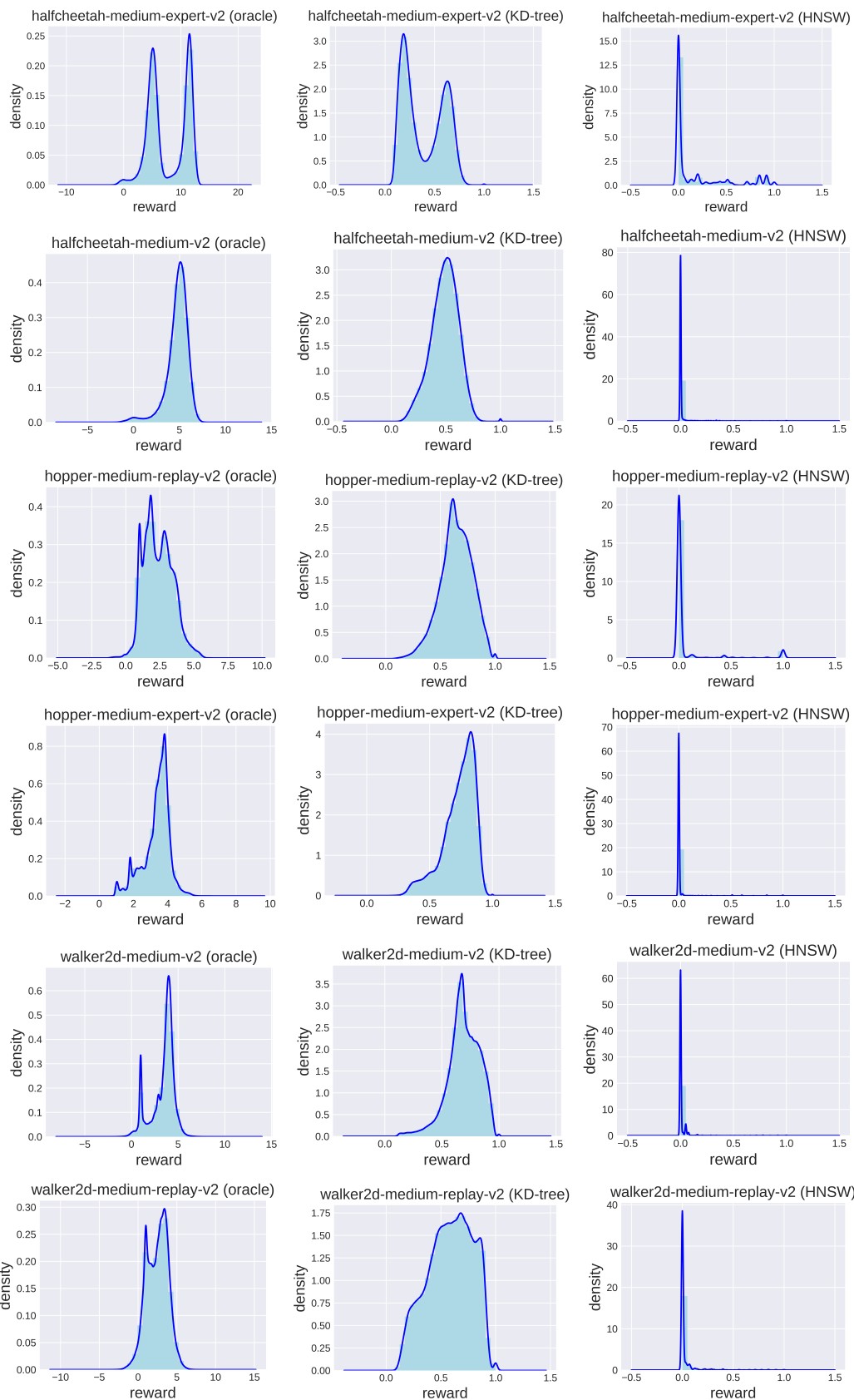

Figure 12: **Density plot comparison of ground-truth rewards and rewards acquired by different search algorithms.** The right two columns show reward distributions of two SEABO variants.

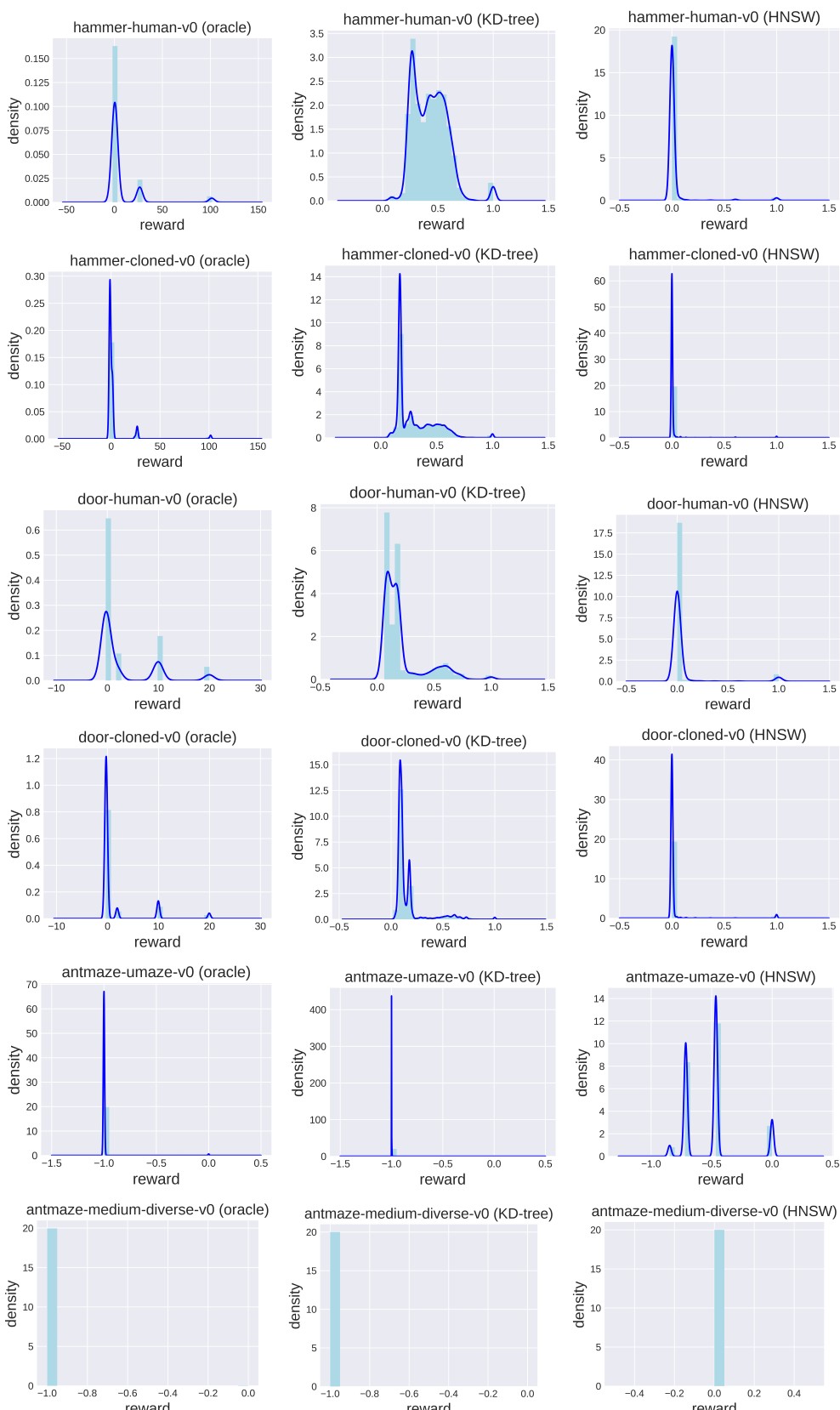

Figure 13: **Density plot comparison of ground-truth rewards and rewards acquired by different search algorithms.** The results are on selected datasets from Adroit and AntMaze tasks.

## G  DISCUSSIONS ON SEABO AND ILR

There are some previous studies that use nearest neighbor-based methods for imitation learning, *e.g.*, Pari et al. (2021). Among them, the most relevant to our work is Ciosek (2022). In this section, we discuss the connections and differences between our method and prior work, ILR (Ciosek, 2022), which can be summarized below:

- **The motivations are varied.** The practical reward formula in ILR is given by $r = 1 - \min_{(s',a') \in D} d_{l_2}((s,a),(s',a'))^2$, which is a *relaxation* of its theoretical version. There exists a gap between the theory and the resulting reward formula. The authors claim that the relaxation is *an upper bound on the scaled theoretical reward* and interpret $L = \min_{(s',a') \in D} d_{l_2}((s,a),(s',a'))^2$ as the $l_2$-diameter of the state-action space. The primary goal of doing so is to reduce imitation learning to RL with a stationary reward for deterministic experts. However, the motivation of SEABO is that we would like to determine the optimality of the single transition (instead of examining whether the transition comes from the expert trajectory or performing relaxation to the rewards). We assume that the transition is near-optimal if it lies close to the expert trajectory. Hence, we assign a larger reward to the transition if it is close to the expert trajectory and a smaller reward otherwise. Meanwhile, SEABO does not require that the expert is deterministic (and also does not require that the environment is deterministic). We aim to adopt SEABO to annotate unlabeled samples in the dataset and train off-the-shelf offline RL algorithms.

- **The methods are different but connected.** The reward formula adopted in ILR is *a special case* of SEABO with Euclidean distance. SEABO does not interpret $L$ as the diameter of the state-action space. SEABO can adopt $N$ nearest neighbors and use their average distance to compute the reward (ILR simply finds the smallest Euclidean distance between sample $(s,a)$ and the expert trajectory). Meanwhile, SEABO is not restricted to Euclidean distance. Our procedure is, that we first find the nearest neighbor of the query sample, and then utilize some distance measurements (different distance measurements can be used here) to decide the distance between the query sample and its nearest neighbor, and finally get the reward by adopting a squashing function. Furthermore, SEABO strongly relies on the nearest neighbor methods (e.g., KD-Tree), and one can use different types of nearest neighbor algorithms in SEABO, while ILR does not emphasize search algorithms. Note that different search algorithms with different hyperparameter setups can result in different final rewards. For example, in *scipy.spatial.KDTree.query*, setting *eps* larger than 0 enables approximate nearest neighbors search and ensures that the $k$-th returned value is no further than $(1 + eps)$ times the distance to the real $k$-th nearest neighbor. This may incur different results from ILR even under Euclidean distance. Moreover, SEABO can also work in state-only regimes, which is both a more general and challenging setting, while ILR strongly relies on the assumption that state-action pairs are present in the expert trajectory in its theory and practical implementation. Finally, one can query with $(s, a, s')$, $(s, a)$ or $(s, s')$ in SEABO (ILR is limited to $(s, a)$), and SEABO adopts a different choice of squashing function.

- **The settings are varied.** SEABO is targeted at the offline imitation learning setting while ILR addresses the online setting. It also turns out that the experiment setup (e.g., number of expert trajectories) is different between SEABO and ILR.

