# OpenReview forum: "SEABO: A Simple Search-Based Method for Offline Imitation Learning"
_ICLR.cc/2024/Conference — ICLR 2024 poster_

### Official Review · Reviewer_LUNW · 2023-10-24

**Soundness:** 4 excellent
**Presentation:** 4 excellent
**Contribution:** 2 fair
**Rating:** 8
**Confidence:** 4

**Summary:**

In this paper, the authors introduce an offline imitation learning algorithm called SEBO, which is centered on the task of learning a reward function from a dataset of expert demonstrations and applying it to unlabeled data to facilitate offline reinforcement learning.

The key innovation in SEBO lies in its use of a straightforward metric for generating the reward function, without the use of any neural networks. The algorithm employs search algorithms to determine the reward function. Specifically, SEBO constructs a KD-tree based on the expert demonstrations. For each unlabeled data point, the algorithm queries the KD-tree to identify its closest neighbor in the expert dataset and assesses the proximity between them. If the distance is minimal (indicating similarity to the expert trajectory), a high reward is assigned; conversely, if the distance is substantial (indicating deviation from the expert trajectory), a low reward is assigned.

The paper evaluates the SEBO algorithm across a range of MuJoCo environments and demonstrates its performance.

**Strengths:**

1. The paper is well written, and the algorithm is studied well on different MuJoCo environments. The authors also conduct a sensitivity analysis with respect to the two parameters $\alpha$ and $\beta$
2. SEBO is efficient, and easy to implement. It costs a minimal overhead over existing Offline RL algorithms, and does not involve training of a new neural network.
3. The authors also evaluate it in a scenario where there is only access to observations, a case that might be of real world importance.

**Weaknesses:**

**Evaluations restricted to deterministic environments**

All evaluations performed in this paper are conducted on deterministic environments (transitions in MuJoCo are completely deterministic), the claim that just one expert trajectory is sufficient might not be true if the environments are stochastic. For example, you may have a good (s,a,s’) pair in the dataset, but you may assign a lower reward to it as its not present in the expert demonstration.  I think there is an inherent correlation between the number of expert trajectories needed and the stochasticity of the environment.

**The paper lacks theoretical justifications, which makes understanding some parts a little difficult. For instance,**

This method might not work in situations where there are multiple ways to solve the same task.
Consider the following example of a navigation problem, where the goal is to navigate to the destination from start position, and there are two ways to solve this task. One that goes left and reaches the goal, the other that goes right and reaches the goal.  Suppose the expert demonstrations takes the left path, and the unlabeled demonstrations (say from an expert policy as well) are from the right path, SEBO will assign low rewards to all transitions and not learn a good policy while BC will probably work.

**Questions:**

1. Is there an inherent assumption being made between the distribution of state-action pairs in the expert demonstrations and the unlabeled dataset?
2. Why are the experiments on D4RL restricted to medium level task and not conducted on expert and random versions of it?
3. Does the stochasticity of environments affect the performance of SEBO?
4. When using point wise matching to determine the reward are you making some inherent assumptions on the transition kernel?

---

> ### Author Response · Authors · 2023-11-17
> **Author Responses to Reviewer LUNW (part 1)**
>
> We thank the reviewer for the thoughtful reviews. We appreciate that the reviewer thinks that our paper is well-written and our method is simple yet effective. Below we try to address the concerns from the reviewer. If we are able to resolve some concerns, we hope that the reviewer will be willing to raise the score.
>
> **Concern 1: evaluations restricted to deterministic environments**
>
> We agree that our evaluations are conducted in deterministic environments. We respectfully argue that most offline IL/offline RL algorithms are evaluated on deterministic environments, and the dynamics in many scenarios (e.g., robotic manipulation) are completely deterministic. Meanwhile, we do not claim that our method is all-around and can handle all scenarios (it is difficult for one single algorithm to do so). With only one expert trajectory, SEABO may fail to produce satisfying rewards for learning in stochastic environments. However, we do not think the *potential* incapability of handling stochastic environments downgrades the contribution of our work.
>
> We agree that, intuitively, there may exist an inherent correlation between the number of expert trajectories needed and the stochasticity of the environment. We deem that it is interesting to check whether the stochasticity of environments affects the performance of SEABO. Nevertheless, as far as we can tell, there are few standard, widely adopted benchmarks with stochastic continuous control environments. Some prior work (e.g., stochastic D4RL Mujoco suite in Appendix C.2 in [1]) introduce stochasticity to the environment by modifying the reward in the dataset, which is somewhat incompatible with SEABO since there are no rewards in the dataset in our setting. We then introduce stochasticity by injecting Gaussian noises (with mean 0 and std $\sigma$) to observations during evaluation, following prior work [2]. We try std=0.05 and std=0.1, respectively, and conduct experiments on MuJoCo datasets. We use IQL as the base algorithm and follow the same hyperparameter setup of SEABO in MuJoCo tasks. We summarize the results below.
>
> | Task Name | IQL (oracle, std=0.05) | IQL+SEABO (std=0.05) | IQL (oracle, std=0.1) | IQL+SEABO (std=0.1) |
> | ---- | :---: | :---: | :---: | :---: |
> | halfcheetah-medium-v2 | 40.9$\pm$1.1 | 40.6$\pm$0.6 | 31.8$\pm$0.8 | 30.4$\pm$0.2 |
> | hopper-medium-v2 | 45.2$\pm$1.8 | 52.4$\pm$3.1 | 33.5$\pm$2.6 | 31.1$\pm$2.8 |
> | walker2d-medium-v2 | 76.8$\pm$4.0 | 80.2$\pm$3.2 | 66.6$\pm$2.2 | 74.3$\pm$5.0 |
> | halfcheetah-medium-replay-v2 | 38.6$\pm$0.9 | 39.5$\pm$1.2 | 30.0$\pm$1.6 | 26.4$\pm$3.1 |
> | hopper-medium-replay-v2 | 51.8$\pm$8.2 | 71.5$\pm$7.9 | 31.7$\pm$7.0 | 33.1$\pm$4.1 |
> | walker2d-medium-replay-v2 | 61.4$\pm$7.5 | 69.2$\pm$4.3 | 51.8$\pm$5.5 | 53.5$\pm$7.3 |
> | halfcheetah-medium-expert-v2 | 36.5$\pm$1.0 | 34.2$\pm$2.9 | 26.8$\pm$1.5 | 24.7$\pm$1.0 |
> | hopper-medium-expert-v2 | 30.4$\pm$5.3 | 35.8$\pm$7.1 | 34.0$\pm$4.8 | 29.5$\pm$3.3 |
> | walker2d-medium-expert-v2 | 101.4$\pm$4.6 | 107.6$\pm$0.7 | 87.1$\pm$9.1 | 90.5$\pm$6.5 |
>
> Table 1. Experimental results in stochastic environments. Oracle means that raw rewards are used for training. The results are averaged over 5 runs.
>
> Based on the results, we find that under stochastic environments, the base algorithm itself fails to behave well (we observe performance collapse on many datasets like halfcheetah-medium-expert-v2). The performance of both IQL and IQL+SEABO decreases with larger noise std. Note that SEABO still outperforms vanilla rewards on many datasets under stochastic environments.
>
> [1] Ma, Y., Jayaraman, D., Bastani, O. Conservative offline distributional reinforcement learning.
>
> [2] Guo, Y., Oh, J., Singh, S., Lee, H. Generative adversarial self-imitation learning.
>
> **Concern 2: the paper lacks theoretical justifications**
>
> We would like to clarify that the main contribution of this paper lies in the empirical side. Many previous offline IL papers like OTR [3] also do not include theoretical analysis. Our concern lies in the fact that theoretical analysis often relies heavily on assumptions that may be hard to satisfy in practice, resulting in a gap between theory and empirical results.
>
> [3] Luo, Y., Jiang, Z., Cohen, S., Grefenstette, E., Deisenroth, M. P. Optimal Transport for Offline Imitation Learning.

---

> > ### Author Response · Authors · 2023-11-17
> > **Author Responses to Reviewer LUNW (part 2)**
> >
> > **Concern 3: SEABO might not work in situations where there are multiple ways to solve the same task**
> >
> > Interesting question! The reviewer is concerned that if there are multiple ways to solve the same task, while all trajectories in the unlabeled dataset take different ways against the expert trajectory, SEABO may allocate low rewards to all transitions and lead to poor training performance. However, under this extreme case, we believe offline IL methods that train discriminator networks may also fail. SEABO still can calculate rewards while the resulting rewards may be of poor quality for task learning. We believe this can be resolved by,
> > - (a) gathering more diversified unlabeled trajectories;
> > - (b) expanding expert trajectories (e.g., if the unlabeled dataset are from the right path and the expert trajectory is from the left path, we can simply involve an expert trajectory from the right path) to involve (all) possible paths;
> > - (c) replacing the expert trajectory with the best trajectory in the unlabeled data (e.g., select the best trajectory from the unlabeled dataset and serve it as the expert trajectory) and then adopting SEABO for acquiring rewards.
> >
> > **Concern 4: Is there an inherent assumption being made between the distribution of state-action pairs in the expert demonstrations and the unlabeled dataset?**
> >
> > SEABO does not involve a distribution matching procedure and hence we believe no inherent assumption is needed. We decide the optimality of the single transition by querying its nearest neighbor in the expert trajectory. A large reward is assigned if their deviation is small, and a smaller reward otherwise. The rewards can be computed anyway. We do not make assumptions on the state-action distribution of offline datasets, and the offline dataset can contain similar trajectories, small amounts of demonstrations or a mixture of trajectories with diversified returns, etc.
> >
> > **Concern 5: results on the random and expert datasets**
> >
> > Sorry for the confusion. We do not involve the results on random and expert datasets as they are extreme cases, i.e., a simple BC agent can learn quite well on expert datasets, and all of the trajectories are of poor quality in random datasets, making it hard to learn a good policy. As the reviewer demands, we include the experimental results of SEABO (with IQL as the base algorithm) on random and expert datasets from MuJoCo. We set reward scale $\alpha=1$, coefficient $\beta=0.5$ for SEABO. We also run IQL with the ground-truth rewards on these datasets. All methods are run for 1M gradient steps and across 5 random seeds. We summarize the final performance below. One can see that SEABO is able to achieve competitive, or even better performance on these datasets as well.
> >
> > | Task name | IQL (oracle) | IQL+SEABO |
> > | ---- | :---: | :---: |
> > | halfcheetah-random-v2 | **13.1$\pm$1.3** | 2.3$\pm$0.1 |
> > | hopper-random-v2 | 7.9$\pm$0.2 | **31.9$\pm$0.1** |
> > | walker2d-random-v2 | **5.4$\pm$1.2** | 5.3$\pm$0.9 |
> > | halfcheetah-expert-v2 | **95.0$\pm$0.5** | 94.4$\pm$0.5 |
> > | hopper-expert-v2 | 109.4$\pm$0.5 | **110.2$\pm$2.2** |
> > | walker2d-expert-v2 | **109.9$\pm$1.2** | **109.9$\pm$0.1** |
> >
> > Table 2. Results on random and expert datasets across 5 different random seeds.
> >
> > **Concern 6: When using point wise matching to determine the reward are you making some inherent assumptions on the transition kernel?**
> >
> > We actually do not make any inherent assumption on the transition kernel during the reward annotation process of SEABO. This is mainly because our method is complete *offline*, indicating that the expert trajectory and the unlabeled dataset are fixed, regardless of whether the underlying environment is stochastic or deterministic. Then, we can query the expert trajectory and measure the deviation between the query sample and its nearest neighbor to produce the reward signal. The whole procedure is independent of the transition kernel.
> >
> > Hopefully, these can resolve the concerns. If there is still something unclear, please let us know!

---

> > > ### Author Response · Authors · 2023-11-20
> > > **Following up with Reviewer LUNW**
> > >
> > > Dear Reviewer LUNW, thanks for your time and efforts in making our paper better. Since the author-reviewer discussion period of this venue ends on Nov 22nd, we wonder if you can kindly check our rebuttal and see if our responses mitigate your concerns. We would appreciate it if you could give us some feedback, and we are ready to have further discussions with the reviewer if there is anything unclear.

---

> > > > ### Comment · Reviewer_LUNW · 2023-11-20
> > > > **Addressed all concerns**
> > > >
> > > > I applaud the detailed rebuttal provided by the authors. The authors have addressed my concerns, and I have modified my score accordingly.

---

> > > > > ### Author Response · Authors · 2023-11-20
> > > > > **Thanks for raising the score!**
> > > > >
> > > > > We are glad that the concerns are addressed. Thanks for raising the score to 8!

---

### Official Review · Reviewer_u2Fn · 2023-10-29

**Soundness:** 2 fair
**Presentation:** 3 good
**Contribution:** 3 good
**Rating:** 6
**Confidence:** 5

**Summary:**

This paper proposes the "SEABO" algorithm, which is an imitation learning algorithm that utilizes an expert dataset along with an unlabeled dataset. SEABO annotates the unlabeled dataset with rewards based on the distance between each state and the closest state in the expert dataset, and then runs an offline RL algorithm on the annotated dataset. The authors show improvements on the D4RL benchmark compared to prior baselines.

**Strengths:**

- This paper proposes a very simple algorithm that achieves good performance relative to more complex methods. I think this type of work has good value to the community in that it introduces easy-to-reproduce results and discourages over-engineering of methods.

- The paper is clear in presentation and mostly well written.

**Weaknesses:**

- There is only empirical analysis of the proposed method. I believe there are certain tasks where SEABO would perform poorly, such as a cliff-walking type of task where there is a precise boundary between what is accetable and what is a failure.

- I hypothesize that the approach will also only work in lower dimensional control environments. This is because the method relies heavily on a  distance function, and this could suffer from the curse of dimensionality in more complex environments.

Minor:
Search in the context of RL and planning typically has a slightly different connotation, which is using some type of tree-based or trajectory-shooting method that optimizes some cost function. In this work search is only used to find states close to an expert. I'm not sure what can be done with this in the writing, but I was expecting a method in the former category after reading the abstract. It may be better to replace "search" with "nearest neighbors" to be more specific.

**Questions:**

"we hypothesize that the transition is near-optimal if it lies close to the expert trajectory" -> Is this strategy always good or does it have weaknesses? I would imagine that certain environments with discontinuities or discrete events (e.g. a car crash) would not favor this strategy.

There is another simple baseline commonly used in offline RL, which is sometimes referred to as percent-BC, which is to imitate the top N% of trajectories in the offline dataset (such as N=10 or N=25). As it is somewhat similar in spirit to SEABO, it would be good to see comparisons to this approach.

Is there any effect of the "curse of dimensionality" for higher dimensional states?

---

> ### Author Response · Authors · 2023-11-17
> **Author Responses to Reviewer u2Fn (part 1)**
>
> We thank the reviewer for the constructive comments. We provide point-to-point clarification to the concerns of the reviewer. Please check our responses below.
>
> **Concern 1: there are only empirical analysis and there are certain tasks where SEABO would perform poorly**
>
> We have to admit that our paper mainly lies on the empirical side, mainly due to the fact that theoretical analysis often incurs a gap between theory and practical implementation or empirical results, and some assumptions made for the benefit of theoretical derivation may not necessarily hold in practice. We agree that a theoretical analysis can be a nice addition to our work, and can leave that for future exploration.
>
> It is interesting to explore how our method behaves when there is a precise boundary between what is acceptable and what is a failure. To that end, we use the risky pointmass environment from [1] (please see the environment code in https://github.com/JasonMa2016/CODAC/blob/main/env/risky_pointmass.py and the details of risky pointmass environment in Appendix C.1 of the CODAC paper) and make some slight modifications. In the original pointmass environment, the agent gets a penalty with some probability. To make the boundary more precise, we directly assign the agent a penalty if it steps into the risky region (line 107-111). That is,
> ```
> if not self.is_safe(self.state):
>     u = np.random.uniform(0, 1)
>     if u > self.risk_prob:
>         cost = 1
>         reward -= self.risk_penalty
> ```
> is modified to
> ```
> if not self.is_safe(self.state):
>     cost = 1
>     reward -= self.risk_penalty
> ```
>
> Other codes are kept unchanged. We collect the offline dataset by training an SAC agent for 1M interactions. We choose the trajectory with the highest return as the expert trajectory. We train an IQL agent upon the collected dataset, and set the reward scale $\alpha=1$ and coefficient $\beta=0.5$ for SEABO. As the environment is simple, we train the IQL agent for 500K gradient steps (hyperparameters are set to be identical to MuJoCo tasks) and summarize the experimental results below. In the risky pointmass environment, some of the data in the expert trajectory can be close to the risky region, while it turns out that IQL+SEABO still beats IQL with vanilla reward.
>
>
> | Task name | $\mu\_{\rm{max}}$ | $\mu\_{\rm{min}}$ | IQL (oracle) | IQL+SEABO |
> | ---- | :---: | :---: | :---: | :---: |
> | risky pointmass | -1.1| -11083.3 | -149.3$\pm$62.1 | **-98.3$\pm$13.2** |
>
> Table 1. Results of SEABO in risky pointmass across 5 different random seeds. $\mu\_{\rm{max}}$ denotes the highest return in the dataset and $\mu\_{\rm{min}}$ denotes the smallest return in the dataset.
>
> Nevertheless, we agree that under some extreme cases (e.g., the expert trajectory is very close to the risky boundary), our method may fail. We actually do not expect that our method can handle all possible situations (and it is hard to do so). If one is concerned with the existence of a precise boundary of failure, one can add some rules to post-process the rewards calculated by SEABO (e.g., penalize those query states that are too close to the boundary).
>
>
> [1] Ma, Y., Jayaraman, D., Bastani, O. Conservative offline distributional reinforcement learning.
>
> **Concern 2: SEABO will only work in lower dimensional control environments**
>
> It is true that MuJoCo tasks are low-dimensional control tasks. However, in the main text, we also evaluate SEABO on Adroit tasks, which controls a 24-DoF simulated Shadow Hand robot to accomplish different tasks like hammering a nail. The observation dimension of `pen-human-v0` gives 45, and the observation dimension of `hammer-human-v0` gives 46. We further include experiments on kitchen datasets, whose observation space gives 60 (which we think can be regarded as high dimensional control tasks), and the results are shown below. Based on the results in Adroit tasks (Table 3 in the paper) and kitchen tasks, we believe SEABO can also work in high dimensional control environments (the curse of dimensionality does not seem to be an issue here).
>
> | Task name | BC | CQL | IQL (oracle) | IQL+SEABO |
> | ---- | :---: | :---: | :---: | :---: |
> | kitchen-complete-v0 | 65.0 | 43.8 | 62.5 | **67.5$\pm$4.2** |
> | kitchen-partial-v0 | 38.0 | 49.8 | 46.3 | **71.0$\pm$4.1** |
> | kitchen-mixed-v0 | 51.5 | 51.0 | 51.0 | **55.0$\pm$3.5**|
> | average score | 51.5 | 48.2 | 53.3 | **64.5** |
>
> Table 2. Results on kitchen environments across 5 different random seeds.
>
> However, it may be challenging to apply SEABO on pixel-based tasks (which are of high dimension and SEABO may suffer from the curse of dimensionality), which we leave as future work since we mainly focus on state-based tasks in this paper. As we stated in Appendix E, one possible solution is to leverage a pre-trained encoder to convert the high-dimensional input into low-dimensional representations. Then, we can apply SEABO to acquire rewards.

---

> > ### Author Response · Authors · 2023-11-17
> > **Author Responses to Reviewer u2Fn (part 2)**
> >
> > **Concern 3: on the hypothesis**
> >
> > We agree that our hypothesis may have limitations under some certain environments. As we stated above, we do not expect our method to handle all conditions. Under those cases, we can add some rules to manually correct the reward output from SEABO (e.g., penalize those states that are too close to the failure boundary). This concern shares some similarities with **Concern 1**, please refer to our responses above.
> >
> > **Concern 4: comparison against percent-BC**
> >
> > Thanks for the advice. In the main text, we have included 10\%-BC as a baseline in Table 2 of the manuscript. It is easy to find that SEABO outperforms 10\%-BC on many datasets.
> >
> > **Concern 5: on the minor issue**
> >
> > We understand the concern. We think that it is okay that we name our method as search-based method, though we merely search for the nearest neighbor. Will the reviewer be satisfied if we include some clarifications (e.g., footnote) in the paper to clarify this? Our proposed method SEABO is the abbreviation of SEArch-Based Offline Imitation Learning. We will need to rename the algorithm if we replace "search" with "nearest neighbors".
> >
> > Hopefully, these can resolve the concerns. If there is still something unclear, please let us know!

---

> > > ### Author Response · Authors · 2023-11-20
> > > **Following up with Reviewer u2Fn**
> > >
> > > Dear Reviewer u2Fn, thanks for your helpful review. As the author-reviewer discussion period is near its end, we wonder if our rebuttal addresses your concerns. Please let us know if there is anything unclear!

---

> > > > ### Comment · Reviewer_u2Fn · 2023-11-22
> > > > **Thank you for the response.**
> > > >
> > > > Dear Authors,
> > > >
> > > > Thank you for the clarifications. On one hand I appreciate the improved experimental results, but also the same limitations that have been highlighted by the other reviewers (lack of a rigorous theoretical analysis, difficulty lack of scaling the distance metric into higher dimensional spaces). I think a score of weak accept is fair for this work, so I will be maintaining my score.

---

> > > > > ### Author Response · Authors · 2023-11-22
> > > > > **Thank you**
> > > > >
> > > > > Thanks for your feedback and positive rating of our work!

---

### Official Review · Reviewer_m3fB · 2023-10-30

**Soundness:** 4 excellent
**Presentation:** 4 excellent
**Contribution:** 3 good
**Rating:** 8
**Confidence:** 4

**Summary:**

The paper proses an approach for imitation learning using nearest-neighbor-based reward computation and offline RL. Given a dataset of unlabeled environment interactions and a single demonstration, they use a KD-tree to compute the distance of each transition to it's nearest neighbor in the demo (euclidean distance) and use this as a pseudo-reward which they can optimize with offline RL. The method is evaluated extensively in D4RL locomotion (and few manipulation) tasks and shows strong performance over prior works.

**Strengths:**

The idea is simple and well-explained in the paper. The empirical validation is thorough, with results across many D4RL locomotion tasks and a small number of manipulation tasks. A representative set of baselines is used and the method shows strong empirical performance. The simple proposed method beats more complex alternatives that need to use optimal transport etc.

**Weaknesses:**

I don't have many issues with the paper. The method has some limitations (see below), but I don't think this invalidates the contributions of the current paper.

One minor weakness is that the approach is mostly evaluated on locomotion tasks for which precision is not the most critical. It could be great to evaluate it on a challenging, long-horizon manipulation task to test the limits of the method. For example the IKEA Furniture assembly benchmark could provide a nice test bed and I would be curious how well the proposed method performs.

Another minor point is that the paper lacks explanation how the euclidean distance is computed on the transition tuple. Do you compute euclidean distance on state, action and next state separately and then sum them? is there a weighting? Providing some details on this would be great!


### Commentary on Limitations

The proposed method seems to have two limitations:

(1) it's unclear how well it would work with visual observations where computing distances and nearest neighbors is a lot more challenging -- the authors mention this as a direction for future work and I agree that this is a limitation shared by many similar works so I wouldn't hold it against this paper. It should be mentioned though that the proposed approach with it's reliance on an analytic distance metric my be more troublesome in such scenarios than e.g. methods that use discriminator-based "distances".

(2) it's unclear how well this would work in tasks that require very precise manipulation, since in such scenarios the proposed method would still assign high rewards to states that are only "nearly" demonstration states but may fail to perform the task in practice. It may be that methods that match trajectory segments instead of individual transitions fare better here?

**Questions:**

--

# Post Rebuttal Comments

Thank you for answering my review.

I appreciate the new experimental results -- performance on Kitchen seems strong and should be included in the paper.

Regarding the furniture assembly benchmark: if offline imitation algorithms struggle in this benchmark, would it be feasible to apply your algorithm in an online context? Also, note that there is a new version of the benchmark (IKEA Furniture Bench, https://clvrai.github.io/furniture-bench/) -- while its focus is on real-world manipulation, it also comes with a simulated counterpart and offline dataset -- you could check that one out as well!

In any case, thank you for adding the experiments and I maintain my recommendation of acceptance. I also skimmed the other reviews and it seems that all reviewers are in agreement that the paper should be accepted.

**Details Of Ethics Concerns:**

--

---

> ### Author Response · Authors · 2023-11-17
> **Author Responses to Reviewer m3fB (part 1)**
>
> We thank the reviewer for the insightful review. We appreciate that the reviewer thinks that our empirical validation is thorough and the performance of SEABO is strong. Please find our clarifications to the concerns below.
>
> **Concern 1: evaluation on a challenging, long-horizon manipulation task to test the limits of the method**
>
> We thank the reviewer for recommending the IKEA Furniture assembly benchmark [1]. We think it is interesting to check how well our method behaves under manipulation tasks that require high precision. We adopt the skill-chaining [2] repository (https://github.com/clvrai/skill-chaining) from the IKEA benchmark author for running experiments, which contains the furniture benchmark. We implement the IQL algorithm on top of `sac_agent.py` in the `robot_learning` repository. We set the expectile=0.7, and tempreature=3.0 in IQL (default values for MuJoCo tasks). We set `control_type=ik` for demonstrate generation and RL training. We use the provided `furniture_sawyer_gen.py` script to generate 200 demonstrations. Though this benchmark supports IL algorithms, it seems to be prepared for *online IL algorithms*. Since no offline datasets are available, we train an SAC agent online for 1M steps and store its collected trajectories. We acquire the offline dataset by combining the demonstration data and the online interaction data. We randomly choose one demonstration from the generated demonstrations as the expert demonstration. We train the IQL agent offline for 1M gradient steps. Due to the fact that the whole procedure is time-consuming (collect online data, generate demonstration, train offline RL algorithm, evaluate offline policy), we only conduct experiments on two tasks from the IKEA benchmark, `chair_ingolf_0650` and `three_blocks`. We set `--num_connects=4, --max_episode_steps=800` for `chair_ingolf_0650` and `--num_connects=2, --max_episode_steps=400` for `three_blocks`. For SEABO, we set reward scale $\alpha=1.0$, coefficient $\beta=0.5$ for these tasks. We summarize the results below (as we are not familiar with this benchmark, we report some results based on the test logging information). We take the results of BC from [1].
>
> |  | BC | IQL (oracle) | IQL+SEABO |
> | ---- | :---: | :---: | :---: |
> | `chair_ingolf_0650` (phase completion) | 1.0 | 1.0$\pm$0.0 | 1.0$\pm$0.0 |
> | `chair_ingolf_0650` (episode return) | - | -5099.7$\pm$371.4 | -4864.2$\pm$768.9 |
> | `three_blocks` (phase completion) | 1.0 | 1.0$\pm$0.0 | 1.0$\pm$0.0 |
> | `three_blocks` (episode return) | - | -3385.6$\pm$701.7 | -3346.18$\pm$679.1 |
>
> Table 1. Results in IKEA furniture benchmark over 5 runs.
>
> [1] Lee, Y., Hu, E. S., Lim, J. J. IKEA furniture assembly environment for long-horizon complex manipulation tasks.
>
> [2] Lee, Y., Lim, J. J., Anandkumar, A., Zhu, Y. Adversarial skill chaining for long-horizon robot manipulation via terminal state regularization.
>
> Note that this benchmark is quite challenging, and requires very high precision manipulation. It turns out that behavior cloning methods generally fail (please see Table 1 in [1]), and IQL with vanilla rewards also fails, indicating that *the base algorithm counts in this benchmark*. Online methods like skill-chaining [2] can handle these tasks well, while it remains unclear whether offline RL algorithms can achieve meaningful performance in this benchmark, which we believe lie out of the scope of our paper.
>
> We further examine how SEABO behaves on some less challenging long-horizon manipulation tasks. To that end, we evaluate SEABO in Kitchen datasets from the D4RL paper. The kitchen environment consists of a 9 DoF Franka robot interacting with a kitchen scene that includes an openable microwave, four turnable oven burners, an oven light switch, a freely movable kettle, two hinged cabinets, and a sliding cabinet door. In kitchen, the robot may need to manipulate different components, e.g., it may need to open the microwave, move the kettle, turn on the light, and slide open the cabinet (precision is required). Please see details of the kitchen environment in [3,4].
>
> [3] Gupta, A., Kumar, V., Lynch, C., Levine, S., Hausman, K. Relay policy learning: Solving long-horizon tasks via imitation and reinforcement learning.
>
> [4] Fu, J., Kumar, A., Nachum, O., Tucker, G., Levine, S. D4rl: Datasets for deep data-driven reinforcement learning.

---

> > ### Author Response · Authors · 2023-11-17
> > **Author Responses to Reviewer m3fB (part 2)**
> >
> > **part1 continued**
> >
> > We run IQL+SEABO on three kitchen datasets using the author-recommended hyperparameters of IQL on the kitchen environment. We set reward scale $\alpha=1$, coefficient $\beta=0.5$ for SEABO. We compare IQL+SEABO against some baselines taken from the IQL paper and summarize the results below. We find that SEABO exhibits superior performance, surpassing IQL with raw rewards by **21.0\%**. We believe these results show that SEABO can aid some long-horizon manipulation tasks.
> >
> > | Task name | BC | CQL | IQL (oracle) | IQL+SEABO |
> > | ---- | :---: | :---: | :---: | :---: |
> > | kitchen-complete-v0 | 65.0 | 43.8 | 62.5 | **67.5$\pm$4.2** |
> > | kitchen-partial-v0 | 38.0 | 49.8 | 46.3 | **71.0$\pm$4.1** |
> > | kitchen-mixed-v0 | 51.5 | 51.0 | 51.0 | **55.0$\pm$3.5**|
> > | average score | 51.5 | 48.2 | 53.3 | **64.5** |
> >
> > Table 2. Results on kitchen environments across 5 different random seeds.
> >
> >
> > **Concern 2: lack explanation of how the Euclidean distance is computed on the transition tuple**
> >
> > Sorry for the confusion. Yes, we simply compute Euclidean distance on state, action, and next state separately and then sum them up. No weighting or other tricks are involved. In practice, for a query transition, we concatenate its state, action, and next state into a single array and directly compute Euclidean distance with its nearest neighbor in the expert trajectory.
> >
> > **Concern 3: commentary on limitations**
> >
> > Thanks for the additional comments on the limitation part.
> >
> > - For visual observations, as stated in Appendix E, it is challenging to directly query the nearest neighbor and acquire rewards with SEABO. A possible solution is to leverage pre-trained image encoders to produce low-dimensional representations. Then, we can query nearest neighbors and compute rewards in the representation space with SEABO. As we mainly focus on state-based tasks in this paper, we leave it as future work. It is possible that discriminator-based methods may produce better rewards in such scenarios.
> >
> > - We include experimental results on the IKEA benchmark and kitchen environment above. Both of them are long-horizon manipulation tasks and require precise manipulation. Our results show that SEABO aids baseline methods on kitchen datasets while not seeming to have advantages over baselines on IKEA tasks (and match trajectory segments instead of individual transitions in SEABO do not bring better performance here).
> >
> > Hopefully, these can resolve the concerns. If there is still something unclear, please let us know!

---

> > > ### Author Response · Authors · 2023-11-20
> > > **Following up with Reviewer m3fB**
> > >
> > > Dear Reviewer m3fB, thanks for your thoughtful review. It would be great if you could give us some comments on the new experiments. We can add the requested experiments concerning the IKEA Furniture assembly benchmark in our revision if the reviewer deems it necessary, and discuss the ability of SEABO to handle long-horizon tasks.

---

### Official Review · Reviewer_KA4c · 2023-10-30

**Soundness:** 3 good
**Presentation:** 3 good
**Contribution:** 3 good
**Rating:** 6
**Confidence:** 4

**Summary:**

This paper proposes a method for Offline Imitation Learning (IL) that defines a reward function based on the Euclidean distance to the nearest neighbor expert state. The method, called SEABO, uses a KD-tree to efficiently query expert states and compute rewards for all transitions. The resulting problem can then be optimized using an arbitrary offline RL algorithm. The experimental results demonstrated improved performance in several tasks of the D4RL benchmark.

**Strengths:**

1. The proposed approach is both novel and simple, and its implementation is efficient due to the use of a KD-tree, without the need for training an extra discriminator.
2. This paper focuses on the context of single-expert-demonstration IL tasks, which is an area of growing interest in the field.
3. I find the discussion on using different search algorithms in Section 5.4 and Appendix Section C interesting.
4. The Limitations section in the appendix is highly appreciated, as it provides valuable guidance on tuning hyperparameters and applying SEABO on visual input.

**Weaknesses:**

1. I'm concerned about the use of Euclidean distance and would suggest that the authors include references justifying the use of this distance metric. This is crucial because there might be scenarios where states that are close in Euclidean distance are, in fact, far apart when accounting for the transitions within the Markov Decision Process (MDP). This particular challenge doesn't arise in discriminator-based methods, mainly due to the use of an additional neural network during training.
2. I believe that "(oracle)" should be omitted from Table 1-3 and 6. For instance, consider "IQL (oracle)": it utilizes the ground truth reward but doesn't rely on expert demonstrations. Removing the ground truth reward and integrating an additional expert demonstration does not necessarily make the task more challenging.
3. The experiments currently compare with only two Offline RL methods (IQL and TD3_BC). It would be better to include more recent baselines such as Trajectory Transformer [[1]], Diffuser [[2]], or other methods.

[1]: https://arxiv.org/abs/2106.02039
[2]: https://arxiv.org/abs/2205.09991

**Questions:**

1. Can SEABO utilize alternative distance metrics in place of the Euclidean distance? If so, how much modification is required?
2. The comparison between ground-truth rewards and the rewards obtained by SEABO (as in Figures 2 and 12) is intriguing. Have you also conducted similar reward comparisons in additional environments, such as AntMaze-v0 and Adroit-v0?

---

> ### Author Response · Authors · 2023-11-17
> **Author Responses to Reviewer KA4c (part 1)**
>
> We thank the reviewer for acknowledging that our method is simple yet effective. Below we try to address the concerns of the reviewer. If we are able to resolve some concerns, we hope that the reviewer will be willing to raise the score.
>
> **Concern 1: on the use of Euclidean distance**
>
> Thanks for the advice. Section 3.2 ***Reward Engineering*** paragraph in [1] lists several works that leverage Euclidean distance for reward engineering, which we think can be a nice reference here. We have included this paper in our revised manuscript (Section 4) for validating the choice of Euclidean distance. The reviewer is concerned that there may exist some scenarios where states that are close in Euclidean distance are, in fact, far apart when accounting for the transitions within the MDP. Based on our experiments, we do not find such phenomena (Euclidean distance can incur good results on most of the evaluated tasks). Suppose that there exist some cases where such a phenomenon occurs, we would suggest the user adopt another distance measurement that can better measure their deviation within the MDP.
>
> [1] Torabi, F., Warnell, G., Stone, P. Recent advances in imitation learning from observation
>
> It is interesting to check how SEABO behaves with other distance measurements. We choose Manhattan distance ($D(x,y) = \sum\_{i}|x\_i - y\_i|$) and cosine distance ($D(x,y) = \dfrac{x\cdot y}{\||x\|| \||y\||}$) for evaluation and summarize the results below. Note that it is quite easy to replace Euclidean distance with other distance measurements (the modification is minor as we only need to replace the function that calculates Euclidean distance with another distance measurement function). Note that we keep hyperparameters of SEABO and IQL unchanged on the evaluated datasets across different distance measurements.
>
> | Task Name | IQL (orcale) | +SEABO(Euclidean) | +SEABO(Manhattan) | +SEABO(cosine) |
> | ---- | :---: | :---: | :---: | :---: |
> | halfcheetah-medium-v2 | **47.4$\pm$0.2** | 44.8$\pm$0.3 | 44.5$\pm$0.1 | 42.6$\pm$0.1 |
> | hopper-medium-v2 | 66.2$\pm$5.7 | **80.9$\pm$3.2** | 74.9$\pm$2.7 | 80.9$\pm$1.1 |
> | walker2d-medium-v2 | 78.3$\pm$8.7 | 80.9$\pm$0.6 | **81.1$\pm$0.7** | 77.3$\pm$0.7 |
> | halfcheetah-medium-replay-v2 | **44.2$\pm$1.2** | 42.3$\pm$0.1 | 42.1$\pm$0.8 | 36.6$\pm$1.3 |
> | hopper-medium-replay-v2 | 94.7$\pm$8.6 | 92.7$\pm$2.9 | **96.7$\pm$3.4** | 69.3$\pm$4.8 |
> | walker2d-medium-replay-v2 | 73.8$\pm$7.1 | **74.0$\pm$2.7** | 61.6$\pm$22.0 | 63.0$\pm$8.4 |
> | halfcheetah-medium-expert-v2 | 86.7$\pm$5.3 | 89.3$\pm$2.5 | **92.0$\pm$0.4** | 71.9$\pm$6.0 |
> | hopper-medium-expert-v2 | 91.5$\pm$14.3 | 97.5$\pm$5.8 | **98.0$\pm$10.7** | 92.7$\pm$10.1 |
> | walker2d-medium-expert-v2 | 109.6$\pm$1.0 | **110.9$\pm$0.2** | 109.1$\pm$0.5 | 105.7$\pm$5.2 |
> | pen-human-v0 | 70.7$\pm$8.6 | 94.3$\pm$12.0 | **96.6$\pm$5.4** | 96.0$\pm$13.8 |
> | pen-cloned-v0 | 37.2$\pm$7.3 | 48.7$\pm$15.3 | 52.3$\pm$20.7 | **52.6$\pm$17.7** |
> | door-human-v0 | 3.3$\pm$1.3 | **5.1$\pm$2.0** | 5.0$\pm$3.5 | 2.7$\pm$0.9 |
> | door-cloned-v0 | **1.6$\pm$0.5** | 0.4$\pm$0.8 | 0.0$\pm$0.0 | 0.0$\pm$0.1 |
>
> Table 1. Comparison of SEABO under different distance measurements. The results are averaged over 5 different random seeds.
>
> It can be seen that Manhattan distance can also incur good performance on some datasets, while cosine distance seems to have unsatisfying performance on numerous datasets. The optimal distance measurement for different dataset may vary, while simply using Euclidean distance can already ensure a good performance.
>
> **Concern 2: (oracle) should be omitted from Table 1-3 and 6**
>
> By specifying *oracle*, we would like to emphasize that the results are obtained by running the offline RL algorithms on datasets with ground-truth rewards. We can remove these if the reviewer deems it necessary.

---

> > ### Author Response · Authors · 2023-11-17
> > **Author Responses to Reviewer KA4c (part 2)**
> >
> > **Concern 3: results on other offline RL methods**
> >
> > To see the performance of SEABO upon some recent baselines, we include two additional baselines, Decision Transformer [2], and TD7 [3]. These methods are chosen due to their simplicity, good reproducibility, and high training efficiency. We use their official GitHub implementations. We adopt the identical hyperparameter setup as TD3\_BC+SEABO for all of the evaluated tasks, and it should not be difficult to reproduce our reported results below. We take the results of DT and TD7 with vanilla rewards from their original papers and summarize the results in Table 2. It can be seen that DT+SEABO achieves competitive or even better performance on numerous datasets. We also find that the performance of TD7+SEABO on many datasets is on par with TD7 trained on raw rewards. We believe the evidence indicates that our method can also work when built upon recent strong methods.
> >
> > | Task Name | DT (oracle) | DT+SEABO | TD7 (oracle) | TD7+SEABO |
> > | ---- | :---: | :---: | :---: | :---: |
> > | halfcheetah-medium-v2 | 42.6$\pm$0.1 | 42.4$\pm$0.4 | 58.0$\pm$0.4 | 47.1$\pm$0.1 |
> > | hopper-medium-v2 | 67.6$\pm$1.0 | 66.5$\pm$5.0 | 76.1$\pm$5.1 | **85.1$\pm$7.3** |
> > | walker2d-medium-v2 | 74.0$\pm$1.4 | **74.1$\pm$4.7** | 91.1$\pm$7.8 | 79.0$\pm$0.4 |
> > | halfcheetah-medium-replay-v2 | 36.6$\pm$0.8 | **37.4$\pm$0.6** | 53.8$\pm$0.8 | 53.6$\pm$0.4 |
> > | hopper-medium-replay-v2 | 82.7$\pm$7.0 | 51.2$\pm$18.3 | 91.1$\pm$8.0 | **98.0$\pm$0.6** |
> > | walker2d-medium-replay-v2 | 66.6$\pm$3.0 | 50.6$\pm$15.6 | 89.7$\pm$4.7 | 70.1$\pm$16.8 |
> > | halfcheetah-medium-expert-v2 | 86.8$\pm$1.3 | **89.8$\pm$0.6** | 104.6$\pm$1.6 | **105.9$\pm$1.8** |
> > | hopper-medium-expert-v2 | 107.6$\pm$1.8 | 94.7$\pm$8.2 | 108.2$\pm$4.8 | 104.7$\pm$0.7 |
> > | walker2d-medium-expert-v2 | 108.1$\pm$0.2 | 107.8$\pm$0.5 | 111.8$\pm$0.6 | 110.5$\pm$0.3 |
> >
> > Table 2.Results of SEABO upon other offline RL baselines across 5 runs. We bold the cell if SEABO outperforms the baseline.
> >
> > [2] Chen, L., Lu, K., Rajeswaran, A., Lee, K., etc. Decision transformer: Reinforcement learning via sequence modeling. NeurIPS 2021.
> >
> > [3] Fujimoto, S., Chang, W. D., Smith, E. etc. For SALE: State-Action Representation Learning for Deep Reinforcement Learning. NeurIPS 2023.
> >
> > **Concern 4: reward comparisons in additional environments, such as AntMaze-v0 and Adroit-v0**
> >
> > We include additional reward comparison between ground-truth rewards and the rewards acquired by using SEABO on hammer-human-v0, hammer-cloned-v0, door-human-v0, door-cloned-v0, and antmaze-umaze-v0, antmaze-medium-diverse-v0. Please see Appendix F in our updated manuscript for details. We find that the reward plots of SEABO resemble those of the vanilla reward signals.
> >
> > Hopefully, these can resolve the concerns. If there is still something unclear, please let us know!

---

> > > ### Author Response · Authors · 2023-11-20
> > > **Following up with Reviewer KA4c**
> > >
> > > Dear Reviewer KA4c, thanks for your time in reviewing our work. As the author-reviewer discussion period is expected to end on Nov 22nd, we wonder if you can kindly check whether our rebuttal resolves your concerns.
> > >
> > > We are more than happy to have further discussions with the reviewer if there are any remaining issues.

---

> > > > ### Comment · Reviewer_KA4c · 2023-11-23
> > > >
> > > > Dear Authors,
> > > >
> > > > Thank you for the detailed responses and the additional experimental results.
> > > >
> > > > The main weakness of this paper is the lack of theoretical justifications. Specifically, the reliance on Euclidean/Manhattan/cosine distances may be inadequate in certain instances of MDP, a concern also raised by other reviewers, including Limitation (2) by Reviewer m3fB, Question (1) by Reviewer u2Fn, and Question (4) by Reviewer LUNW.
> > > >
> > > > I also wish to reiterate that the use of "(oracle)" might be misleading. Offline RL algorithms trained with ground-truth rewards do not incorporate the additional single-expert demonstration and, therefore, cannot be considered true oracles. The true oracle would require modifying the offline RL algorithms to somehow utilize the additional expert demonstration.
> > > >
> > > > Nevertheless, as acknowledged in the author responses to Reviewer LUNW, the authors do not claim that this method is universally applicable and capable of addressing all scenarios. I find the empirical perspective of this work interesting, demonstrating that simple methods can yield good performance. The inclusion of new experimental results has addressed some of my initial concerns, leading me to revise my score accordingly.

---

> > > > > ### Author Response · Authors · 2023-11-23
> > > > > **Thanks for raising the score!**
> > > > >
> > > > > We thank the reviewer for raising the score to 6! We will remove the use of *(orcale)* in our next revision, before the end of the rebuttal. Thanks for your time and efforts in making our paper better.

---

> > > > > ### Author Response · Authors · 2023-11-23
> > > > > **We have uploaded a new revision**
> > > > >
> > > > > Dear Reviewer KA4c, we have uploaded a new revision and removed *(oracle)* from all of the tables in the main text and the appendix. We also revised the corresponding descriptions in the table captions and texts. Thank you for your comments and feedback.

---

### Comment · Area_Chair_jhwA · 2023-11-22
**From AC.**

Hi Authors!

Thank you for the detailed responses to the reviews. I have a slight concern about novelty. Would it be possible for you to relate your method to the heuristic proposed on page 8 of [1]? While [1] is about online RL, I think the method is very related to what you are doing.  For the avoidance of doubt, you do not have to cite the paper [1] if you do not think it is relevant. In that case, please just write a comment outlining how you think the approaches are different.

[1] K.Ciosek Imitation Learning by Reinforcement Learning

---

> ### Author Response · Authors · 2023-11-22
> **Quick Response to AC**
>
> Dear AC, thanks for your comments. We appreciate it that you recommend [1]. We have checked this paper and would respectfully argue that SEABO is different from [1].
>
> **On the motivation**
>
> If we understand it correctly, the motivation of acquiring intrinsic reward in [1] is to check whether the state-action pair comes from expert dataset, i.e., $R=\\{(s,a)\in D\\}$. The theoretical analysis is constructed based on such reward formula. Its practical form gives $R_{int}=1-\min_{(s^\prime,a^\prime)\in D}d_{l2}((s,a), (s^\prime,a^\prime))^2$, which incurs some gap from its theory. The authors write that such relaxation can be seen as *an upper bound on the scaled theoretical reward* and $\min_{(s^\prime,a^\prime)\in D}d_{l2}((s,a), (s^\prime,a^\prime))^2$ denotes the $L_2$-diameter of the state-action space.
>
> However, our motivation in this paper is quite straightforward, that we would like to determine the optimality of the single transition (instead of examining whether the transition comes from the expert trajectory or performing relaxation to the rewards). We assume that the transition is near-optimal if it lies close to the expert trajectory. Hence, we assign larger rewards to the transition if it is close to the expert trajectory and a smaller reward otherwise.
>
> **On the method**
>
> There are also many differences between the two methods,
>
> - [1] measures the diameter of state-action space, which gives $\min_{(s^\prime,a^\prime)\in D}d_{l2}((s,a), (s^\prime,a^\prime))^2$. This is actually *a special case* of SEABO with Euclidean distance. However, SEABO is not restricted to Euclidean distance. Our procedure is, we first find the nearest neighbor of the query sample, and then utilize some distance measurement (different distance measurement can be used here) to decide the distance between the query sample and its nearest neighbor, and finally get the reward by adopting a squashing function. One can adopt any distance measurement to get the final reward (please see our responses to Reviewer KA4c).
>
> - SEABO strongly relies on the nearest neighbor methods (please check our main paper), and one can use different types of nearest neighbor algorithms in SEABO, while [1] does not emphasize search algorithms. Note that different search algorithms with different hyperparameter setup can result in different final rewards.
>
> - SEABO can also work in state-only regimes, while [1] strongly relies on the assumption that state-action pairs are present in the expert trajectory in its theory and practical implementation
>
> **On the setting**
>
> As AC comments, SEABO is targeted at the offline imitation learning setting while [1] addresses the online imitation learning setting. Meanwhile, it seems that the experimental setup of our paper and [1] also differs.
>
> Please let us know if we miss something, or if AC is satisfied with our response. We are more than happy to update our manuscript in the next revision to include the recommended paper and discuss the differences between our work and [1]. We would upload the final revision before the end of the rebuttal.
>
> [1] K.Ciosek Imitation Learning by Reinforcement Learning

---

> > ### Author Response · Authors · 2023-11-23
> > **Following up with AC**
> >
> > Dear AC, we have uploaded a new revision and added a new section in the appendix (please check Appendix G) to clarify the connections and differences between our work and [1], as promised.
> >
> > Please refer to our revised manuscript for details. We would like to highlight some additional points here.
> >
> > - [1] relies on the assumption that the expert is deterministic, while SEABO does not make any assumption on whether the expert policy is deterministic
> >
> > - SEABO does not interpret $L$ as the diameter of the state-action space and can compute reward with $N$ nearest neighbors, while [1] merely finds the smallest Euclidean distance
> >
> > - One can query with $(s,a,s^\prime)$, $(s,a)$ or $(s,s^\prime)$ in SEABO, while [1] is limited to $(s,a)$
> >
> > - SEABO leverages a squashing function instead of directly using the resulting distance
> >
> > Please let us know if you have remaining concerns. We are ready to have further discussions with AC if possible.
> >
> > [1] K.Ciosek Imitation Learning by Reinforcement Learning

---

### Meta-Review · Area_Chair_jhwA · 2023-12-06

**Metareview:**

The paper attempts to address the offline imitation learning problem where we have (i) a (small) dataset of expert data and (ii) a (typically much larger) dataset of exploration data. The method is a heuristic which rewards the agent for coming close to a state from the expert dataset.

Strengths:
- strong empirical performance
- well written and understandable paper
- well written limitations section

Weaknesses:
- no theory
- experiments don't use rliable (https://github.com/google-research/rliable) or similar

**Justification For Why Not Higher Score:**

Paper is purely empirical but empirical evaluation is not the best possible (no performance profiles etc).

**Justification For Why Not Lower Score:**

A simple method wins against more complicated baselines. This deserves publication.

---

### Decision · Program_Chairs · 2024-01-16

Accept (poster)